# Loss of Ena/VASP interferes with lamellipodium architecture, motility and integrin-dependent adhesion

**Julia Damiano-Guercio[1], Laëtitia Kurzawa[2,3†], Jan Mueller[4†], Georgi Dimchev[5†], Matthias Schaks[5,6], Maria Nemethova[4], Thomas Pokrant[1], Stefan Brühmann[1], Joern Linkner[1], Laurent Blanchoin[2,3], Michael Sixt[4], Klemens Rottner[5,6], Jan Faix[1]\***

[1]Institute for Biophysical Chemistry, Hannover Medical School, Hannover, Germany; [2]CytoMorphoLab, Laboratoire de Physiologie cellulaire et Végétale, Interdisciplinary ResearchInstitute of Grenoble, CEA, CNRS, INRA, Grenoble-Alpes University, Grenoble, France; [3]CytomorphoLab, Hôpital Saint-Louis, Institut Universitaire d'Hematologie, UMRS1160, INSERM/AP-HP/UniversitéParis Diderot, Paris, France; [4]Institute of Science and Technology Austria (IST Austria), Klosterneuburg, Austria; [5]Division of Molecular Cell Biology, Zoological Institute, Technical University Braunschweig, Braunschweig, Germany; [6]Molecular Cell Biology Group, Helmholtz Centre for Infection Research, Braunschweig, Germany

**\*For correspondence:**
faix.jan@mh-hannover.de

[†]These authors contributed equally to this work

**Competing interests:** The authors declare that no competing interests exist.

**Abstract** Cell migration entails networks and bundles of actin filaments termed lamellipodia and microspikes or filopodia, respectively, as well as focal adhesions, all of which recruit Ena/VASP family members hitherto thought to antagonize efficient cell motility. However, we find these proteins to act as positive regulators of migration in different murine cell lines. CRISPR/Cas9-mediated loss of Ena/VASP proteins reduced lamellipodial actin assembly and perturbed lamellipodial architecture, as evidenced by changed network geometry as well as reduction of filament length and number that was accompanied by abnormal Arp2/3 complex and heterodimeric capping protein accumulation. Loss of Ena/VASP function also abolished the formation of microspikes normally embedded in lamellipodia, but not of filopodia capable of emanating without lamellipodia. Ena/VASP-deficiency also impaired integrin-mediated adhesion accompanied by reduced traction forces exerted through these structures. Our data thus uncover novel Ena/VASP functions of these actin polymerases that are fully consistent with their promotion of cell migration.

## Introduction

Adhesion and migration are invariably driven by continuous and dynamic actin cytoskeleton remodeling (*Blanchoin et al., 2014*). The major protrusive structure of migrating cells on flat and rigid substrata is the sheet-like lamellipodium (*Letort et al., 2015*; *Rottner et al., 2017*). The extension of dense and branched actin filaments in the lamellipodium drives its growth and pushes the membrane forward (*Koestler et al., 2008*; *Mullins et al., 1998*; *Pollard and Borisy, 2003*; *Vinzenz et al., 2012*). Branching by Actin-related protein (Arp) 2/3 complex is activated by the WAVE regulatory complex (WRC) downstream of Rac subfamily GTPase signaling (*Eden et al., 2002*; *Ismail et al., 2009*; *Molinie and Gautreau, 2018*). Consistently, knockdown or knockout of essential Arp2/3 complex (*Suraneni et al., 2012*; *Wu et al., 2012*) or WRC subunits (*Innocenti et al., 2004*; *Schaks et al., 2018*; *Steffen et al., 2004*) or Rac GTPases (*Schaks et al., 2018*; *Steffen et al., 2013*) all abrogated lamellipodium formation. The lateral flow of actin filaments and bundles embedded in the lamellipodium, called microspikes, is driven by actin assembly at the

membrane (*Oldenbourg et al., 2000*). When microspikes transform and protrude beyond the lamellipodium edge, they are called filopodia (*Small, 1988*; *Svitkina et al., 2003*). Yet, in spite of both being built of parallel actin filament bundles and sharing constituents, filopodia and microspikes display important differences. As opposed to microspikes as integral parts of lamellipodial networks, filopodia can kink and bend (*Nemethova et al., 2008*), arising independently and even in excess in the absence of lamellipodia (*Gupton et al., 2005*; *Koestler et al., 2013*; *Steffen et al., 2006*; *Wu et al., 2012*). Consistently, filopodia can form around the entire cell periphery and at the dorsal surface (*Block et al., 2008*; *Bohil et al., 2006*; *Pellegrin and Mellor, 2005*). Moreover, rates of actin polymerization in microspikes and lamellipodia are indistinguishable from each other (*Lai et al., 2008*; *Oldenbourg et al., 2000*), whereas those in filopodia appear to be regulated independently.

Filament elongation is driven by formins and Ena/VASP proteins. Formins are dimeric multi domain proteins that remain tightly associated with growing filament barbed ends while accelerating their growth (*Kovar et al., 2006*). Several formins such as mDia2 (*Block et al., 2008*; *Yang et al., 2007*) and the formin-like family members 2 (FMNL2) and 3 (*Block et al., 2012*; *Harris et al., 2010*; *Kage et al., 2017*; *Young et al., 2018*) localize at lamellipodia and filopodia tips, and have been implicated in driving these protrusions and cell migration.

Ena/VASP family proteins localize to sites of active actin assembly including focal adhesions (FA) (*Gertler et al., 1996*; *Reinhard et al., 1995*), stress fibers (*Gateva et al., 2014*; *Rottner et al., 2001*; *Tojkander et al., 2015*), and lamellipodial or filopodial tips (*Rottner et al., 1999*; *Svitkina et al., 2003*). Vertebrates harbor Vasodilator-stimulated phosphoprotein (VASP), mammalian Protein Enabled Homolog (Mena), and Ena-VASP-like protein (Evl). All family members are tetramers encompassing domains allowing for interactions with FPPPP-containing proteins, actin monomers, profilin-actin complexes and actin filaments (*Bear and Gertler, 2009*). Single Ena/VASP tetramers are poorly processive actin polymerases, but processivity and resistance against heterodimeric capping protein that stops barbed end growth, increases markedly upon oligomerization or clustering (*Breitsprecher et al., 2011a*; *Breitsprecher et al., 2008*; *Brühmann et al., 2017*; *Hansen and Mullins, 2010*).

Consistent with their accumulation at filopodia tips, genetic removal of all three Ena/VASP proteins was previously shown to abrogate filopodia formation in neuronal cells (*Kwiatkowski et al., 2007*). The functions of Ena/VASP proteins in cell motility have remained more controversial. Elimination of the single family member VASP in *Dictyostelium* cells diminishes random motility and chemotaxis (*Han et al., 2002*; *Litschko et al., 2017*) suggesting a stimulatory role VASP on cells migration. Consistently, VASP accumulation at lamellipodia tips was shown to positively correlate with protrusion rates in B16-F1 mouse melanoma cells (*Rottner et al., 1999*) and fish keratocytes (*Lacayo et al., 2007*). On the other hand, genetic inactivation of VASP and Mena or mitochondrial Ena/VASP sequestration in fibroblasts was reported to increase cell migration (*Bear et al., 2002*; *Bear et al., 2000*). This phenotype was explained by lamellipodia protruding more persistently after interference with Ena/VASP function, and containing shorter and more branched filaments. Excess of Ena/VASP, in contrast, generated lamellipodia with longer, less branched filaments, prone to lifting rearwards during membrane ruffling, therefore driving migration less efficiently (*Bear et al., 2002*). However, a similar approach yielded opposite results on the migratory behavior of *Drosophila* haemocytes with slower migration upon Ena sequestration and faster migration rates upon Ena overexpression (*Tucker et al., 2011*). The latter observations again resemble the increased motility described for breast cancer cells overexpressing Mena (*Philippar et al., 2008*). Taken together, we are in need of a more generalizable view on Ena/VASP function in cell migration. Despite described controversies concerning motility, the proposed positive regulatory function of Ena/VASP on lamellipodial filaments was consistent with their clear relevance in the movement through the host cell cytosol of *Listeria* harboring the Ena/VASP ligand ActA (*Geese et al., 2002*; *Loisel et al., 1999*; *Skoble et al., 2001*).

Finally, although Ena/VASP proteins are prominent FA components, their roles in cell substrate adhesion and FA formation have also remained controversial. Both inhibitory (*Galler et al., 2006*) and stimulatory roles (*Gupton et al., 2012*; *Kang et al., 2010*; *Puleo et al., 2019*; *Young and Higgs, 2018*) on adhesion were reported, whereas one study did not detect any effect in fibroblasts lacking Mena and VASP (*Bear et al., 2000*). To shed more light on these questions and ease interpretation of all these conflicting observations, we generated somatic gene disruptions of individual, two or all three Ena/VASP family members in distinct, mesenchymal cell types, and explored the

impact of individual and collective Ena/VASP member removal on formation of different protrusion types, efficiency of 2D and 3D migration and adhesion.

## Results

### Loss of Ena/VASP impairs adhesive, 2D cell migration in B16-F1 cells

To evaluate the function of Ena/VASP proteins in cell migration in a comparably fast mesenchymal cell type, we sequentially inactivated the three Ena/VASP paralogues Evl, VASP and Mena using CRISPR/Cas9 technology in B16-F1 mouse melanoma cells (*Supplementary file 1*). Respective protein loss in independent clonal cell lines was confirmed by immunoblotting and verified by sequencing of genomic target sites (*Figure 1A*; *Supplementary file 2*). Migration rates of B16-F1 wild-type and two independent lines of each genotype were then analyzed on laminin by phase-contrast, time-lapse microscopy (*Figure 1—source data 1*). Interestingly, consecutive Ena/VASP member removal caused an increasing phenotype in 2D migration rate, ranging from a modest reduction of 1.25 ± 0.37 µm/min and 1.19 ± 0.47 µm/min for single Evl KO mutants (E-KO) as compared to control (1.44 ± 0.4 µm/min), to 0.62 ± 0.25 µm/min and 0.57 ± 0.27 µm/min in Evl and VASP double-KOs (EV-KO), and 0.47 ± 0.19 µm/min and 0.43 ± 0.22 µm/min in triple-KOs (EVM-KO) (*Figure 1B*, *Figure 1—video 1*). This was accompanied by a gradual increase in directionality culminating in EVM-KO cells that were 46% more directional as compared to wild-type B16-F1 (*Figure 1C*). Quantitative analyses of the turning angles revealed an incrementally increased frequency at 0° from 10.2% in wild-type to 12.3% in single KO, 16.1% in double KO and 16.9% in triple KO cells, in support of the higher directionality in mutant cells that coincided with their reduced motility (*Figure 1D*). Since independently generated Ena/VASP mutant cell lines behaved highly similar in distinct assays, data derived from one triple-KO clone (#23.7.66) are shown below. To ensure that observed phenotypes were Ena/VASP-specific, the same parameters were analyzed in reconstituted EVM-KO mutant cells ectopically expressing EGFP-tagged VASP, Evl or Mena (*Figure 1—figure supplement 1*). Re-expression of VASP, which was found to contribute most efficiently to motility, largely rescued cell speed, directionality and frequency of turning angles (*Figure 1E–F*, *Figure 1—figure supplement 2A*). The rescue experiments with EGFP-tagged Evl or Mena yielded similar results, but these proteins were slightly less effective in restoring cell speed (*Figure 1E–F*). Finally, we calculated the mean square displacement (MSD) in wild-type and the entire collection of mutant cells to assess their effective directional movement. Despite their higher directionality, presumably owing to their considerably slower motility, all Ena/VASP mutant cells displayed incrementally decreasing and lower MSD values as compared to control (*Figure 1G*). Notably, in EVM-KO cells rescued with EGFP-tagged VASP, Evl or Mena, the MSD values were again markedly increased (*Figure 1—figure supplement 2B*). Since Ena/VASP proteins have previously also been implicated in cancer cell invasion (*Philippar et al., 2008*), we additionally analyzed migration performance of confined EVM-KO and B16-F1 control cells in 3D Matrigel invasion assays. Surprisingly, however, at least with the assay conditions employed, we did not find any significant differences in invasion rates between these cell lines (*Figure 1—figure supplement 3*, *Figure 1—video 2*). Future work will have to clarify how this result compares to migration in additional 3D settings, such as through extracellular matrices with variable geometries or pore sizes. However, although the mechanistic details of this result remain to be established, it suggests, at least, that the robust, Ena/VASP-driven promotion of actin-based protrusion is less important for migration in confinement, which particularly relies on actomyosin contractility, as previously reported for other cell types (*Poincloux et al., 2011*; *Ramalingam et al., 2015*).

### Ena/VASP-deficiency diminishes 2D cell migration in fibroblasts

To corroborate our findings in more strongly adherent cell types, we first examined MV[D7] mouse embryonic fibroblasts. This cell line was described previously to derive from Mena/VASP-deficient mice and selected to lack detectable Evl expression (*Bear et al., 2000*) or concluded to contain at best trace amounts (*Auerbuch et al., 2003*). We did confirm the absence of Mena and VASP in these cells by immunoblotting (*Figure 2A*). Surprisingly, however, our newly generated antibodies raised against and affinity-purified with Evl (see Materials and methods) now identified the protein at low, but clearly detectable levels in this cell line. Thus, to obtain fibroblasts devoid of all three Ena/VASP

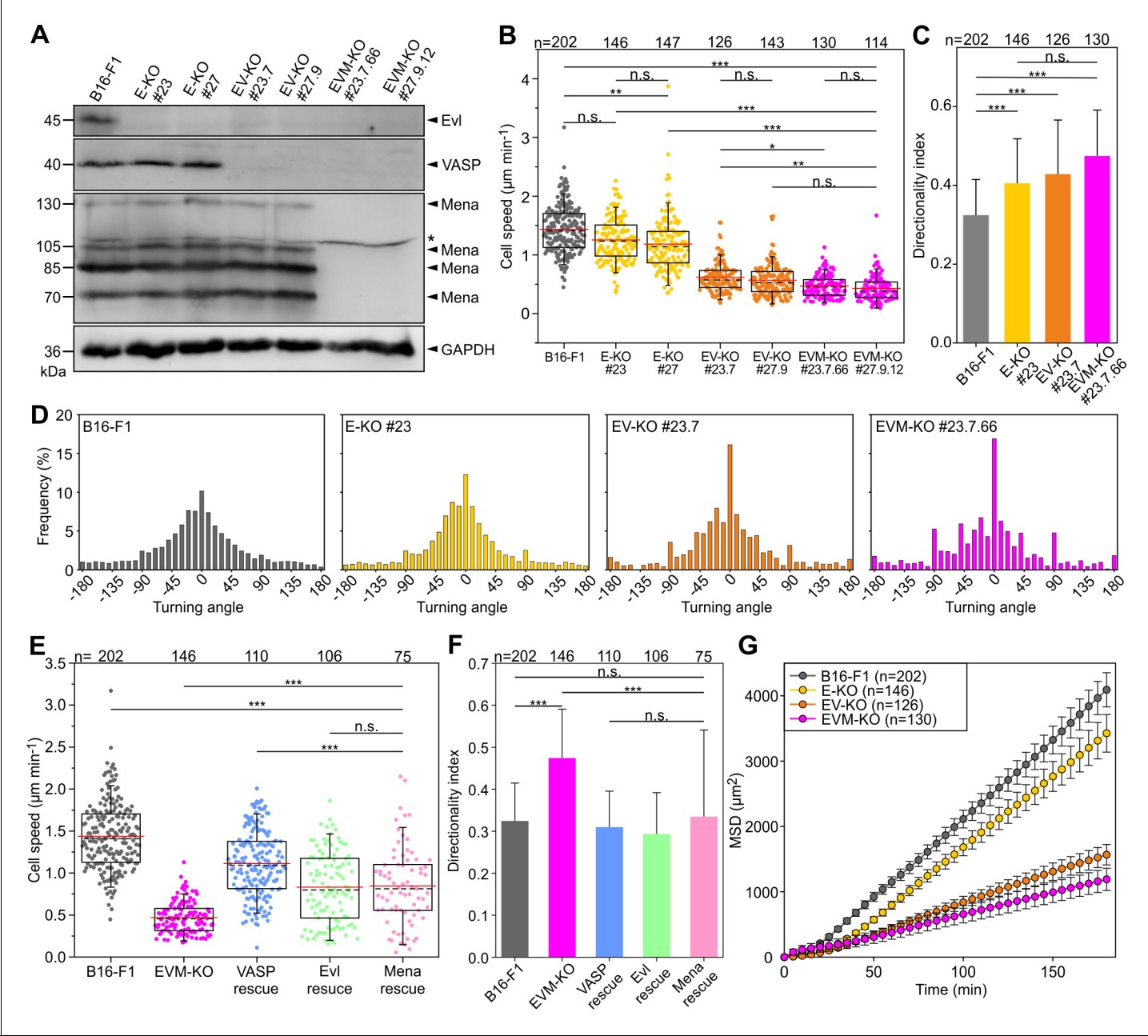

**Figure 1.** Loss of Ena/VASP-proteins impairs 2D cell migration in B16-F1 cells. (**A**) To obtain independent triple-knockout mutant cell lines, the two independent Evl single knockout mutants (E-KO #23 and #27) derived from B16-F1 mouse melanoma cells, were separately used to the generate independent Evl/VASP double mutants (E/V-KO #23.7 and #27.9) followed by generation of two individual triple-KO mutants additionally lacking Mena (E/V/M-KO #23.7.66 and #27.9.12). Elimination of Evl, VASP and all Mena isoforms by CRISPR/Cas9 in B16-F1 cells was confirmed by immunoblotting using specific antibodies (asterisk indicates nonspecific band). GAPDH was used as loading control. (**B**) Consecutive gene disruption of the three Ena/VASP paralogues increasingly diminishes cell migration on laminin. At least three time-lapse movies from three independent experiments were analyzed for each cell line. (**C**) Directionality increased with consecutive inactivation of the three Ena/VASP paralogues. Bars represent arithmetic means ± SD. (**D**) Distribution of turning angles during migration of B16-F1 and derived mutant cells. (**E–G**) Cell speed and directionality are largely rescued by ectopic expression of EGFP-tagged VASP, Evl or Mena in EVM-KO #23 cells. (**F**) Bars represent arithmetic means ± SD. (**G**) Analyses of mean square displacement of wild-type *versus* mutant cells. Respective symbols and error bars represent means ± SEM. (**B–E**) The boxes in box plots indicate 50% (25–75%) and whiskers (5–95%) of all measurements, with dashed black lines depicting the medians, arithmetic means are highlighted in red. (**B–C**) and (**E–F**) Non-parametric, Kruskal-Wallis test and Dunn's Multiple Comparison test were used to reveal statistically significant differences between datasets. *p≤0.05, **p≤0.01, ***p≤0.001; n.s.; not significant. n, number of cells analyzed from at least three independent experiments.

The online version of this article includes the following video, source data, and figure supplement(s) for figure 1:

*Figure 1 continued on next page*

*Figure 1 continued*

**Source data 1.** Source data for details of cell migration including cell speeds, directionality indices and MSD values.

**Figure supplement 1.** Expression levels of Ena/VASP proteins in reconstituted EVM-KO cells compared to endogenous levels assessed by immune blotting.

**Figure supplement 2.** Loss of Ena/VASP proteins affects 2D migration and directionality.

**Figure supplement 2—source data 1.** Source data for details of cell migration including turning angles and MSD values *Figure 1—figure supplement 2*.

**Figure supplement 3.** Loss of Ena/VASP-proteins does not affect invasion of B16-F1 cells.

**Figure supplement 3—source data 1.** Source data for details of invasion including invasion speeds *Figure 1—figure supplement 3*.

**Figure 1—video 1.** Loss of Ena/VASP proteins in B16-F1 cells impairs 2D cell migration, related to *Figure 1*.

https://elifesciences.org/articles/55551#fig1video1

**Figure 1—video 2.** Loss of Ena/VASP-proteins does not affect invasion of B16-F1 cells, related to *Figure 1—figure supplement 3*.

https://elifesciences.org/articles/55551#fig1video2

---

proteins, we employed CRISPR/Cas9 to eliminate Evl in $MV^{D7}$ cells, with Evl disruption confirmed by immunoblotting and sequencing as described for B16-F1 cells above (*Figure 2A*, *Supplementary file 2*). Since distinct triple-KO cell lines behaved again in a highly comparable fashion, only data from clone #31 referred to as MVE-KO are shown. We first analyzed 2D migration on fibronectin by phase-contrast time-lapse imaging of $MV^{D7}$ and MVE-KO as well as MVE-KO cells rescued with EGFP-tagged Evl (*Figure 2—source data 1*). In MVE-KO, cell speed (0.41 ± 0.17 µm/min) was significantly reduced as compared to $MV^{D7}$ control (0.56 ± 0.19 µm/min), whereas migration of MVE-KO cells expressing EGFP-Evl (0.68 ± 0.28 µm/min) was markedly increased (*Figure 2B*). Remarkably, in spite of their higher cell speed, $MV^{D7}$ exhibited even lower MSD values than MVE-KO cells (*Figure 2C*), apparently caused by the lower directionality of the former, and as evidenced by their trajectories in radar plots (*Figure 2—figure supplement 1A–B*). As expected, triple-KO cells reconstituted with Evl displayed strongly increased MSD values (*Figure 2C*) and decreased directionality (*Figure 2—figure supplement 1A–B*).

Additionally, we analyzed directional cell migration of these cell lines in wound closure scratch assays (*Figure 2D*). Average wound closure rates were reduced by 24% in MVE-KO cells (18121 ± 2658 $\mu m^2 h^{-1}$) as compared to $MV^{D7}$ cells (23906 ± 2042 $\mu m^2 h^{-1}$) (*Figure 2E*). Interestingly, reconstitution of MVE-KO cells with stable Evl expression significantly increased wound closure to rates virtually identical to $MV^{D7}$ controls (*Figure 2F*). Thus, despite comparably low expression of Evl in $MV^{D7}$ fibroblasts, its elimination caused clearly detectable impairment of cell migration.

To assess the contribution of the two other Ena/VASP family members in fibroblast motility, we also disrupted the *Enah* and *Vasp* genes encoding Mena and VASP individually or in combination in mouse NIH 3T3 fibroblasts by CRISPR/Cas9 technology, and again confirmed elimination of respective proteins by immunoblotting and sequencing (*Figure 2G*, *Supplementary file 2*). Migration rates of NIH 3T3 wild-type and two independent lines of each genotype were then again analyzed on fibronectin by phase-contrast time-lapse imaging. Comparable to the findings in B16-F1 cells, consecutive removal of Mena and VASP in NIH 3T3 cells also caused a stepwise increase in migration phenotype, ranging from a rather modest average reduction of about 10.6% for single Mena KO mutants (M-KO) to 31.6% in Mena and VASP double-KO (MV-KO) mutants as compared to control (*Figure 2H*). Consistently, these Ena/VASP mutant cells also exhibited incrementally decreasing and lower MSD values as compared to NIH 3T3 wild-type controls (*Figure 2I*). All these findings combined, strongly suggest, quite strikingly, that Ena/VASP proteins execute conserved and prominent, positive regulatory functions in adhesive, 2D cell migration, in a fashion irrespective of cell type (melanoma cells and fibroblast alike) and thus type of mesenchymal migration and/or signaling condition, as induced by extracellular matrices (laminin *versus* fibronectin).

To relate the specific contributions of individual Ena/VASP family members to cell migration, we finally determined the cellular concentrations of expressed orthologues in B16-F1, NIH 3T3 and $MV^{D7}$ cells. For this, we titrated defined amounts of recombinant proteins with total cell lysates from a given number of cells in immunoblots, and calculated resulting absolute levels of each protein in the different cell types (*Figure 2J*). Interestingly, in spite of considerably higher expression of Mena in B16-F1 and NIH 3T3 cells as compared to VASP, and in particular relative to Evl, loss of Mena

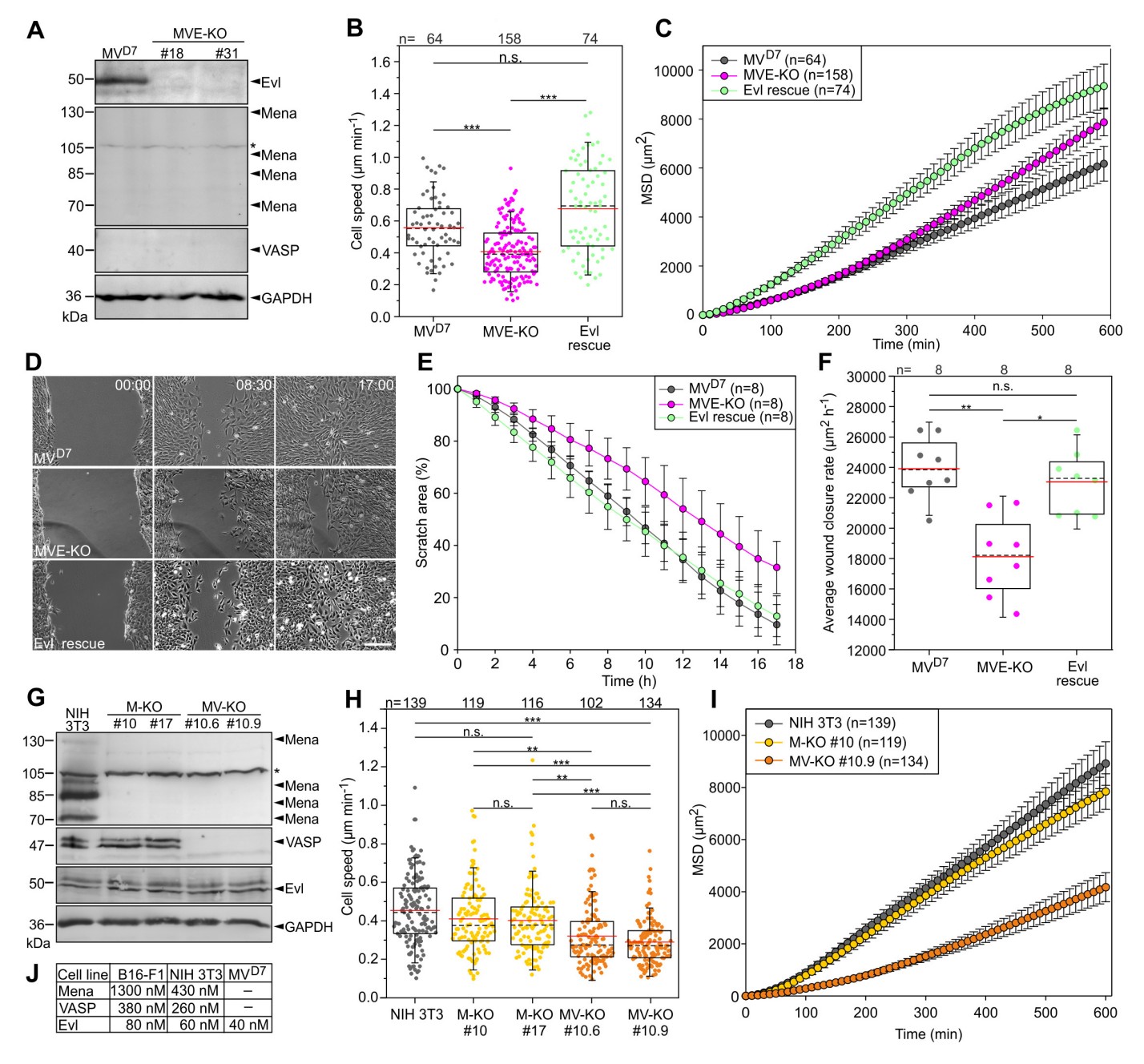

**Figure 2.** Inactivation of Ena/VASP proteins in various fibroblasts impairs 2D cell migration. (**A**) MV^D7 cells lack Mena and VASP, but still express Evl. Elimination of Evl by CRISPR/Cas9 in MV^D7 fibroblasts was confirmed by immunoblotting in independent clonal cell lines (MVE-KO). GAPDH was used as loading control. Asterisk indicates nonspecific band in Mena blot. (**B**) Elimination of Evl in MV^D7 cells decreased cell speed on fibronectin-coated glass and could be rescued by expression of Evl. (**C**) Analyses of mean square displacement of MV^D7, MVE-KO and reconstituted cells as indicated. Data points represent arithmetic means ± SEM. At least three time-lapse movies from three independent experiments were analyzed for each cell line. (**D**) Representative frames from wound healing movies of MV^D7, MVE-KO and reconstituted cells as indicated. MVE-KO cells were not able to close the wound after 17 hr. Bar, 200 μm. (**E**) Reduction of wound area over time. n, number of movies analyzed. Data are means ± SD. (**F**) Average wound closure rate. n, number of movies analyzed. (**G**) Immunoblot of independent single and double-KO mutants derived from NIH 3T3 fibroblasts lacking Mena (M–KO) or Mena and VASP (MV-KO) by CRISPR/Cas9 technology. GAPDH was used as loading control. (**H**) Consistent with findings in B16-F1 cells, consecutive gene disruption of these two Ena/VASP paralogues in NIH 3T3 fibroblasts again increasingly diminished cell migration on fibronectin. (**I**) Analyses of mean square displacement of NIH 3T3, M-KO and MV-KO cells as indicated. Data points represent arithmetic means ± SEM. n, number of cells tracked. At least three time-lapse movies from three independent experiments were analyzed for each cell line. (**J**) Total cytoplasmic concentrations of Ena/VASP proteins in investigated cell lines. (**B, F and H**) Boxes in box plots indicate 50% (25–75%) and whiskers (5–95%) of all

*Figure 2 continued on next page*

*Figure 2 continued*

measurements, with dashed black lines depicting the medians, arithmetic means are highlighted in red. Non-parametric, Kruskal-Wallis test and Dunn's Multiple Comparison test were used to reveal statistically significant differences between datasets. *p≤0.05, **p≤0.01, ***p≤0.001; n.s.; not significant. n, number of cells analyzed from at least three independent experiments.

The online version of this article includes the following source data and figure supplement(s) for figure 2:

**Source data 1.** Source data for details of cell migration including cell speeds, MSD values, wound healing scratch areas and average wound closure rates *Figure 2*.

**Figure supplement 1.** Elimination of Evl in MV^D7 cells affects directionality.

**Figure supplement 1—source data 1.** Source data for details of cell migration including directionality indices *Figure 2—figure supplement 1*.

affected migration much less strongly than, for instance, removal of VASP. This suggests a certain differentiation of Ena/VASP family member function despite clearly overlapping functions.

## Loss of Ena/VASP reduces lamellipodium width and abrogates microspike formation

Since 2D migration on flat surfaces is mainly driven by actin assembly in the lamellipodium, we then analyzed actin filament (F-actin) distribution in B16-F1 and EVM-KO cells after phalloidin staining (*Figure 3—source data 1*). As opposed to B16-F1 control cells, which displayed prominent lamellipodia with numerous microspikes, EVM-KO cells developed strongly compromised lamellipodia largely lacking microspikes (*Figure 3A*, *Figure 3—video 1*). Quantification of F-actin contents in lamellipodia revealed about 45% reduction in the triple mutant, and the average width of lamellipodia was diminished by about 65% to 0.8 ± 0.2 μm when compared to wild-type with 2.3 ± 0.7 μm. In mutant cells reconstituted with EGFP-VASP, F-actin intensity increased even 40% above wild-type levels, and lamellipodium width was readily restored to 2.7 ± 0.8 μm (*Figure 3B–C*). Similar, but less pronounced effects were found after rescue with EGFP-tagged Evl or Mena as compared to VASP (*Figure 3—figure supplement 1A–C*). Given the evident lack of microspikes, we stained for the actin-crosslinking protein fascin, a well-established constituent of microspikes in B16-F1 cells (*Vignjevic et al., 2006*). Again, the triple mutant virtually lacked fascin-containing bundles, as opposed to wild-type and reconstituted mutants expressing VASP, despite unchanged global expression of fascin in the mutant (*Figure 3D–G*). This was corroborated by imaging of wild-type and EVM-KO cells expressing EGFP-fascin (*Figure 3—video 2*). Finally, we examined the gene dose-dependent requirement for Ena/VASP family members in microspike formation and found that virtually complete loss of microspikes was observed only upon disruption of all three family members (*Figure 3H*).

## Potent filopodia inducers fail to rescue microspike formation in Ena/VASP-deficient cells

We then asked whether microspikes can be reformed in the triple-mutant upon transient expression of potent factors known to induce peripheral protrusions, and in particular related protrusive structures such as filopodia (*Block et al., 2008*; *Bohil et al., 2006*; *Kage et al., 2017*; *Pellegrin and Mellor, 2005*). We also used all Ena/VASP members as controls. Evl and Mena both rescued canonical lamellipodia with microspikes in EVM-KO comparable to VASP, with Mena forming microspikes above average length (*Figure 4A*, *Figure 4—figure supplement 1A–B*). Strikingly, none of the agents established previously to be capable of inducing filopodia formation, that is constitutively active variants of the small Rho GTPase Rif (L77), the formins mDia2 (mDia2ΔDAD), FMNL3 (E275) and to lesser extent FMNL2 (E272), or unconventional myosin X were able to restore microspike-containing lamellipodia, indicating that microspike formation requires more specific activities that are critically dependent on Ena/VASP function (*Figure 4A*, *Figure 4—figure supplement 2*).

To explore this potential molecular difference between microspikes and filopodia more directly, we induced filopodia in the absence of lamellipodia in B16-F1 and EVM-KO cells. Since it is commonly assumed that lamellipodia emerge by extension of Arp2/3 complex-generated, dendritic filament networks downstream of WRC and Rac signaling (*Pollard, 2007*; *Schaks et al., 2018*; *Steffen et al., 2013*; *Wu et al., 2012*), we inhibited Arp2/3 complex with high concentrations of CK666 in combination with cell seeding on low concentrations of laminin, which aided inhibitor-

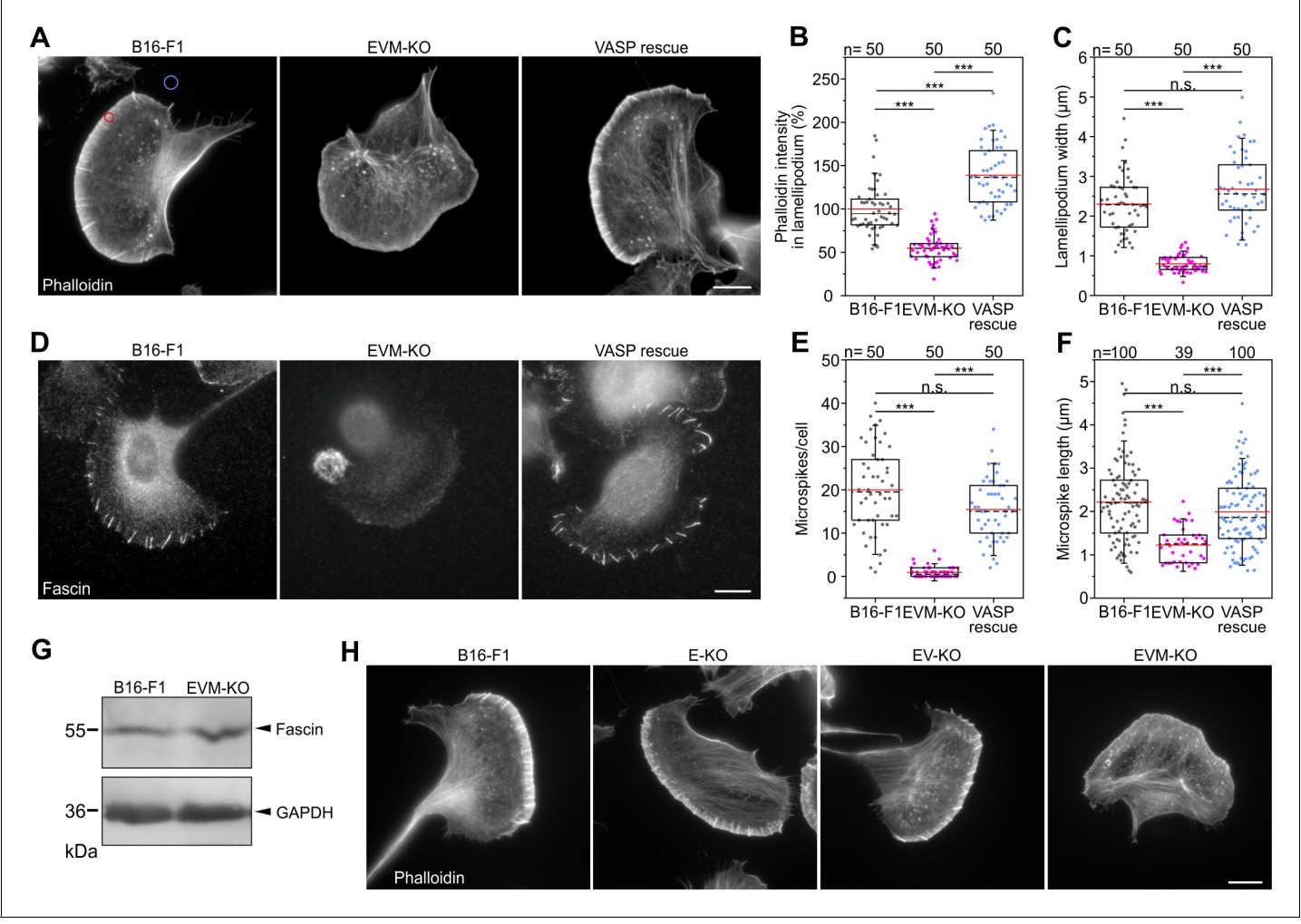

**Figure 3.** Loss of Ena/VASP perturbs lamellipodia and abrogates microspike formation in B16-F1 cells. (**A**) Representative examples of lamellipodia from wild-type B16-F1, EVM-KO and reconstituted EVM-KO cells transiently expressing EGFP-VASP. Cells migrating on laminin and stained for the actin cytoskeleton with phalloidin. Bar, 10 μm. (**B**) Quantification of F-actin intensities in lamellipodia (red circle in A) of wild-type and mutant cells after subtraction of background (blue circle in A). (**C**) Quantification of lamellipodia width in wild-type and mutant cells. (**D**) Loss of Ena/VASP markedly diminishes fascin-decorated microspikes, which are rescued by EGFP-VASP. Immunostaining with fascin antibody. Bar, 10 μm. (**E**) Quantification of microspikes in wild-type and mutant cells. (**F**) Quantification of microspike length in wild-type and mutant cells. (**G**) Comparable expression of fascin in wild-type and EVM-KO cells was confirmed by immunoblotting using fascin-specific antibodies. GAPDH was used as loading control. (**H**) Representative examples of lamellipodia from wild-type B16-F1 cells, single E-KO, double EV-KO and triple EVM-KO mutants. Displayed cells migrating on laminin were fixed and stained with phalloidin to visualize their F-actin cytoskeleton. Bars, 10 μm. Note presence of microspikes in B16-F1 wild-type cells, single E-KO and double EV-KO mutants, but not in the triple EVM-KO mutant cell. (**B–C and E–F**) Boxes in box plots indicate 50% (25–75%) and whiskers (5–95%) of all measurements, with dashed black lines depicting the medians, arithmetic means are highlighted in red. Non-parametric, Kruskal-Wallis test and Dunn's Multiple Comparison test were used to reveal statistically significant differences between datasets. ***p≤0.001; n.s.; not significant. n, number of cells analyzed (**B, C, and E**) or microspikes (**F**) from at least three independent experiments.

The online version of this article includes the following video, source data, and figure supplement(s) for figure 3:

**Source data 1.** Source data for details of lamellipodial actin including relative phalloidin intensities, lamellipodium widths, microspike numbers per cell and microspike length *Figure 3*.

**Figure supplement 1.** Expression of Evl and Mena is sufficient for rescuing lamellipodial parameters in B16-F1 cells.

**Figure supplement 1—source data 1.** Source data for details of lamellipodial actin including relative phalloidin intensities and lamellipodium widths *Figure 3—figure supplement 1*.

**Figure 3—video 1.** Elimination of all three Ena/VASP proteins impairs lamellipodium formation and abolishes microspikes, related to *Figure 3*.
https://elifesciences.org/articles/55351#fig3video1

**Figure 3—video 2.** Elimination of all three Ena/VASP proteins abrogates microspikes, related to *Figure 3*.
https://elifesciences.org/articles/55351#fig3video2

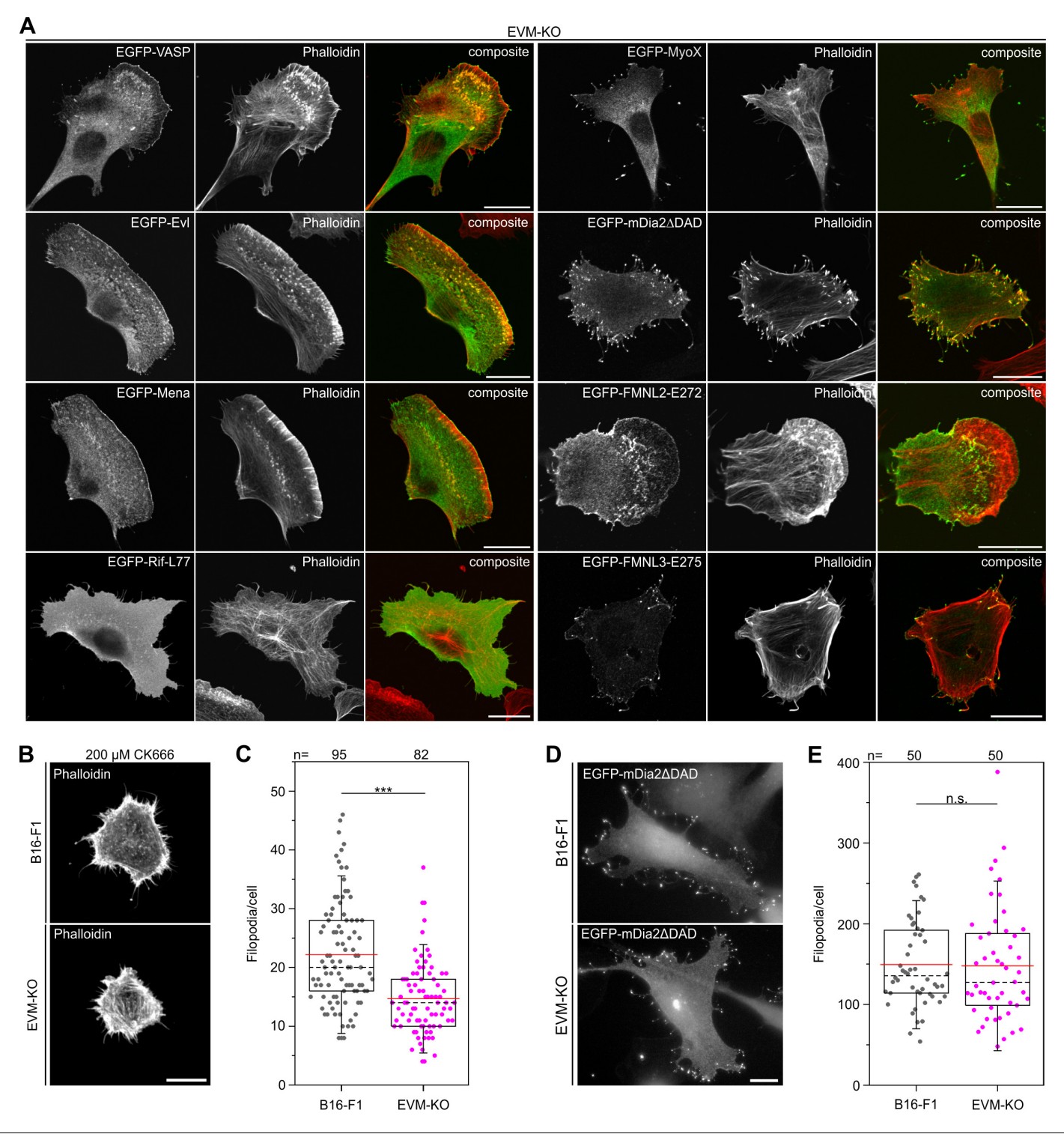

**Figure 4.** Microspike formation is exclusively rescued by all three Ena/VASP members, but not by active Rif-L77, myosin X, active mDia2ΔDAD or active FMNL2 and −3. (**A**) Images display cells stained for the actin cytoskeleton with phalloidin and respective, expressed EGFP-tagged protein as indicated. Bars, 20 μm. (**B**) EVM-KO cells form filopodia in the absence of lamellipodia. Representative examples of phalloidin-stained wild-type B16-F1 and EVM-KO cells devoid of lamellipodia after treatment with 200 μM CK666 seeded at low laminin (1 μg/mL). Bar, 10 μm. (**C**) Quantification of filopodia in CK666-treated B16-F1 and EVM-KO cells. (**D**) Unchanged filopodium formation in B16-F1 and EVM-KO cells triggered by transient expression of active mDia2. Representative examples of B16-F1 and EVM-KO cells after labelling with GFP nanobodies are shown. Bar, 10 μm. (**E**) Quantification of filopodia revealed no difference between wild-type and mutant cells. (**C and E**) Boxes in box plots indicate 50% (25–75%) and whiskers (5–95%) of all

*Figure 4 continued on next page*

*Figure 4 continued*

measurements, with dashed black lines depicting the medians, arithmetic means are highlighted in red. Non-parametric, Mann-Whitney U rank sum test was used to reveal statistically significant differences between datasets. ***p≤0.001; n.s.; not significant. n, number of cells analyzed from at least three independent experiments.

The online version of this article includes the following source data and figure supplement(s) for figure 4:

**Source data 1.** Source data for details of filopodia formation *Figure 4*.
**Figure supplement 1.** Quantification of microspikes after rescue of EVM-KO cells with EGFP-tagged Evl or Mena.
**Figure supplement 1—source data 1.** Source data for details of microspike formation including microspike number per cell and microspike length *Figure 4—figure supplement 1*.
**Figure supplement 2.** Active FMNL2 or −3 do not rescue microspikes in EVM-KO cell.
**Figure supplement 2—source data 1.** Source data for details of microspike formation including microspike number per cell and microspike length *Figure 4—figure supplement 2*.

mediated suppression of lamellipodia formation in these conditions (*Figure 4B*). Analyses of phalloidin-stained cells revealed that albeit reduced in formation frequency per cell, filopodia formation was still possible in EVM-KO cells (66% compared to B16-F1 control on low laminin). This confirmed that Ena/VASP family can indeed contribute to filopodia formation, as previously observed in *D. discoideum* (*Han et al., 2002*). At the same time, these data also illustrated that filopodia can indeed form to significant extent without lamellipodia, for example upon inhibition of Arp2/3 complex and thus in the absence of microspikes (*Figure 4B–C*, *Figure 4—source data 1*), in spite of the reduced frequencies of filopodia formation observed here. Conversely, however, induction of filopodia formation by expression of constitutively active mDia2 (*Block et al., 2008*; *Yang et al., 2007*) was virtually identical in the EVM-KO mutant as compared to B16-F1 cells (*Figure 4D–E*, *Figure 4—source data 1*), supporting the view of multiple pathways leading to filopodium formation (*Young et al., 2015*).

## Loss of Ena/VASP changes distribution of Arp2/3 complex and capping protein and affects lamellipodial dynamics

To explore if or how loss of Ena/VASP function affected Arp2/3 complex accumulation, we immunolabeled the cells for the Arp2/3 complex and found that it accumulated in a peripheral band much narrower than commonly observed in wild-type lamellipodia (reduced by 64%), but with roughly 75% higher intensity at the tips of mutant lamellipodia (*Figure 5A–C*, *Figure 5—source data 1*). The mutant phenotype was again Ena/VASP-specific, since it was perfectly rescued by transient expression of EGFP-tagged VASP and Evl, and at least partly by Mena. More specifically, VASP was slightly more effective than Evl, and Mena was capable of mediating a statistically significant rescue only concerning width of observed Arp2/3 complex signal (*Figure 5A–C*, *Figure 5—figure supplement 1A–C*). We also monitored the distributions of heterodimeric capping protein (CP), the F-actin binding protein cortactin and the WRC-subunit WAVE2. CP and cortactin distributions were highly reminiscent of changes in Arp2/3, again much narrower (reduced by 60% and 67%, respectively), and with approximately 75% and 60% increased intensities in EVM-KO as compared to controls (*Figure 5D–F*, *Figure 5—source data 1*, *Figure 5—figure supplement 1D–I*). Interestingly, we found no noticeable difference in distribution and intensity of WAVE2 in wild-type *versus* EVM-KO (*Figure 5—figure supplement 1J–L*). This together with reversion to or close to wild-type levels of Arp2/3 complex, CP and cortactin by EGFP-tagged Ena/VASP family members illustrated the impact of the latter on the lamellipodial Arp2/3 complex machinery downstream, and thus independent of its major lamellipodial activator, WRC.

Next, we asked whether or to which extent lamellipodia protrusion was affected. To this end, we recorded wild-type, EVM-KO and VASP-reconstituted cells randomly migrating on laminin, and determined respective protrusion rates by kymograph analyses (*Figure 5G–H*, *Figure 5—source data 1*, *Figure 5—video 1*). Quantification revealed lamellipodia protrusion to be reduced by 35% in EVM-KO cells as compared to control or EVM-KO rescued with EGFP-VASP (*Figure 5I*, *Figure 5—source data 1*). Notably, despite reduced protrusion, lamellipodia persistence was at best slightly reduced in EVM-KO cells, but not in a statistically significant fashion (*Figure 5J*), confirming that protrusion effectivity can be readily uncoupled from persistence (*Block et al., 2012*; *Kage et al., 2017*).

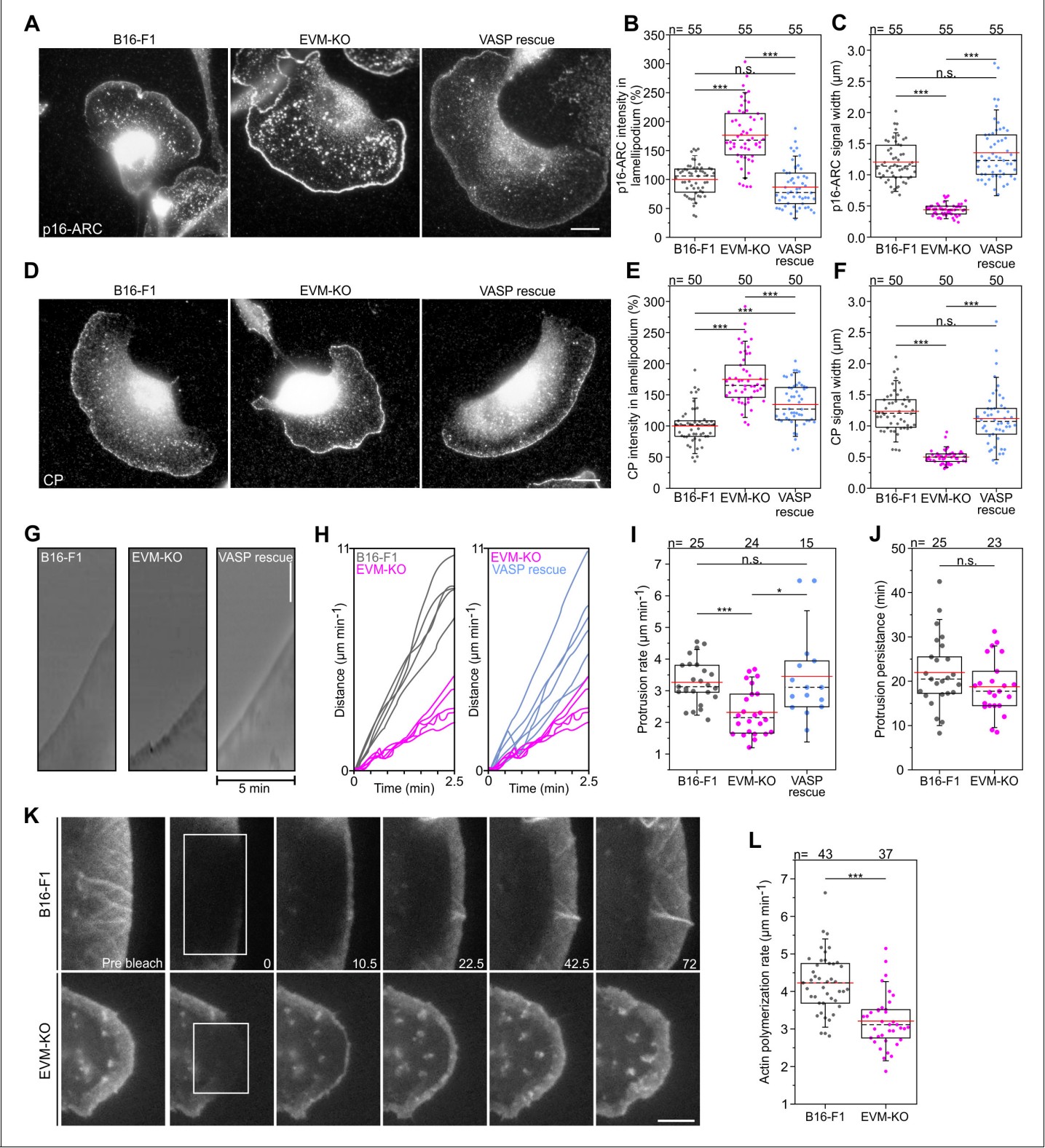

**Figure 5.** Loss of Ena/VASP affects lamellipodial parameters and protrusion dynamics. (**A**) Immunolabelling of the Arp2/3 complex subunit p16-ARC (ArpC5A) in wild-type, mutant and reconstituted cells as indicated. Bar, 10 μm. (**B**) Quantification of Arp2/3 complex intensities in lamellipodia. (**C**) Quantification of Arp2/3 complex signal width. (**D**) Immunolabelling of CP in cell types as indicated. Bar, 10 μm. (**E**) Quantification of CP signal intensities in lamellipodia. (**F**) Quantification of CP signal width in lamellipodia. (**G**) Loss of Ena/VASP reduces the efficiency of lamellipodium protrusion. Kymographs of representative phase-contrast movies are shown. Bar, 10 μm. (**H**) Multiple examples of lamellipodium protrusion in B16-F1 *versus* EVM-

*Figure 5 continued on next page*

*Figure 5 continued*

KO *versus* EVM-KO cells reconstituted with VASP. (**I**) Quantification of protrusion rates. B-C, E-F and I, *p≤0.05, ***p≤0.001 and n.s., not significant by Kruskal-Wallis test and Dunn's Multiple Comparison test. n, number of cells. (**J**) Quantification of protrusion persistence. (**K**) FRAP of EGFP-ß-actin in lamellipodia of B16-F1 and EVM-KO cells bleached as indicated by white rectangles. Numbers in post-bleach images correspond to seconds after bleach. Bar, 3 µm. (**L**) Average actin assembly rates of respective cell types expressing EGFP-ß-actin. Note reduced lamellipodial actin assembly in the mutant. J-L, ***p≤0.001, n.s., not significant by Mann-Whitney U rank sum test. n, number of cells analyzed.

The online version of this article includes the following video, source data, and figure supplement(s) for figure 5:

**Source data 1.** Source data for details of lamellipodial proteins including relative p16-ARC and CP intensities and signal widths, and of protrusions including protrusion rates and persistence, and actin polymerization rates *Figure 5*.

**Figure supplement 1.** Loss of Ena/VASP affects additional lamellipodial parameters.

**Figure supplement 1—source data 1.** Source data for details of lamellipodial proteins including relative p16-ARC, CP, cortactin and WAV2 intensities and signal widths *Figure 5—figure supplement 1*.

**Figure 5—video 1.** Impaired protrusion of EVM-KO cells migrating on laminin, related to *Figure 5*.

https://elifesciences.org/articles/55351#fig5video1

**Figure 5—video 2.** Loss of Ena/VASP in B16-F1 cells reduces actin network polymerization, related to *Figure 5*.

https://elifesciences.org/articles/55351#fig5video2

Ena/VASP proteins are potent actin polymerases (*Breitsprecher et al., 2011a*; *Breitsprecher et al., 2008*; *Hansen and Mullins, 2010*), implying that these factors could affect dynamics of actin assembly in lamellipodia. To test this experimentally, we assessed lamellipodial actin network assembly rates in wild-type and EVM-KO cells (*Figure 5K*, *Figure 5—video 2*). Lamellipodial actin network translocation was also reduced by about 22% in EVM-KO as compared to B16-F1 controls, suggesting that the effects on protrusion described above can be explained, at least in part, by reduced actin network polymerization (*Figure 5L*, *Figure 5—source data 1*).

## Loss of Ena/VASP deteriorates lamellipodium architecture

To gain quantitative insights into the ultrastructural details of lamellipodial architecture at the single filament level, we employed three-dimensional (3D) electron microscopy of negatively stained B16-F1 wild-type and EVM-KO samples, allowing computer-assisted tracing of single filaments in 3D space (*Mueller et al., 2017*; *Winkler et al., 2012*). Consistent with actin assembly in lamellipodia tips (*Lai et al., 2008*), filaments are oriented with their barbed ends facing the membrane (*Narita et al., 2012*). In analogy to recent work (*Mueller et al., 2017*), barbed ends were scored in digitalized tomograms as the ends of filaments proximal to the leading edge and pointed ends as those distal to the leading edge. Barbed ends either represent actively growing or capped filaments, whereas pointed ends designate Arp2/3 complex-generated branch sites, severed or debranched filaments. In contrast to lamellipodia of wild-type cells, EVM-deficient lamellipodia contained much sparser and less organized filament networks (*Figure 6A*, *Figure 6—video 1*), which is consistent with the F-actin stainings in *Figure 3A–B*. Consistently, assessment of actin filaments in the front region (within a ~ 1 µm broad zone behind the edge) revealed considerably shorter filaments in the mutant (*Figure 6B*, *Figure 6—source data 1*). Moreover, toward the interior of the lamellipodium, filament density in the EVM-mutant decreased noticeably faster as compared to control, while at the edge, barbed ends occurred in excess in the mutant (*Figure 6C*, *Figure 6—source data 1*). Consistent with reduced filament lengths (*Figure 6B*) and increased Arp2/3 intensities at the edge of EVM-KO cells (*Figure 5A–B*), assessed pointed end densities were significantly above control values in particular within the first half micron of the edge (*Figure 6C*). Finally, although both control and mutant lamellipodia exhibited a wide array of filament angles abutting the protruding front, with a clear maximum around 90° relative to the edge (*Figure 6D*, *Figure 6—source data 1*), the distribution of filament angles in EVM-deficient lamellipodia was distinct, with the 90° peak being clearly less prominent (17% lower than in control). Conversely, filament fractions with diagonal angles were about doubled compared to controls. These data suggest that reduced filament mass and disordered filament network geometry in the mutant are causative for reduced protrusion and network assembly.

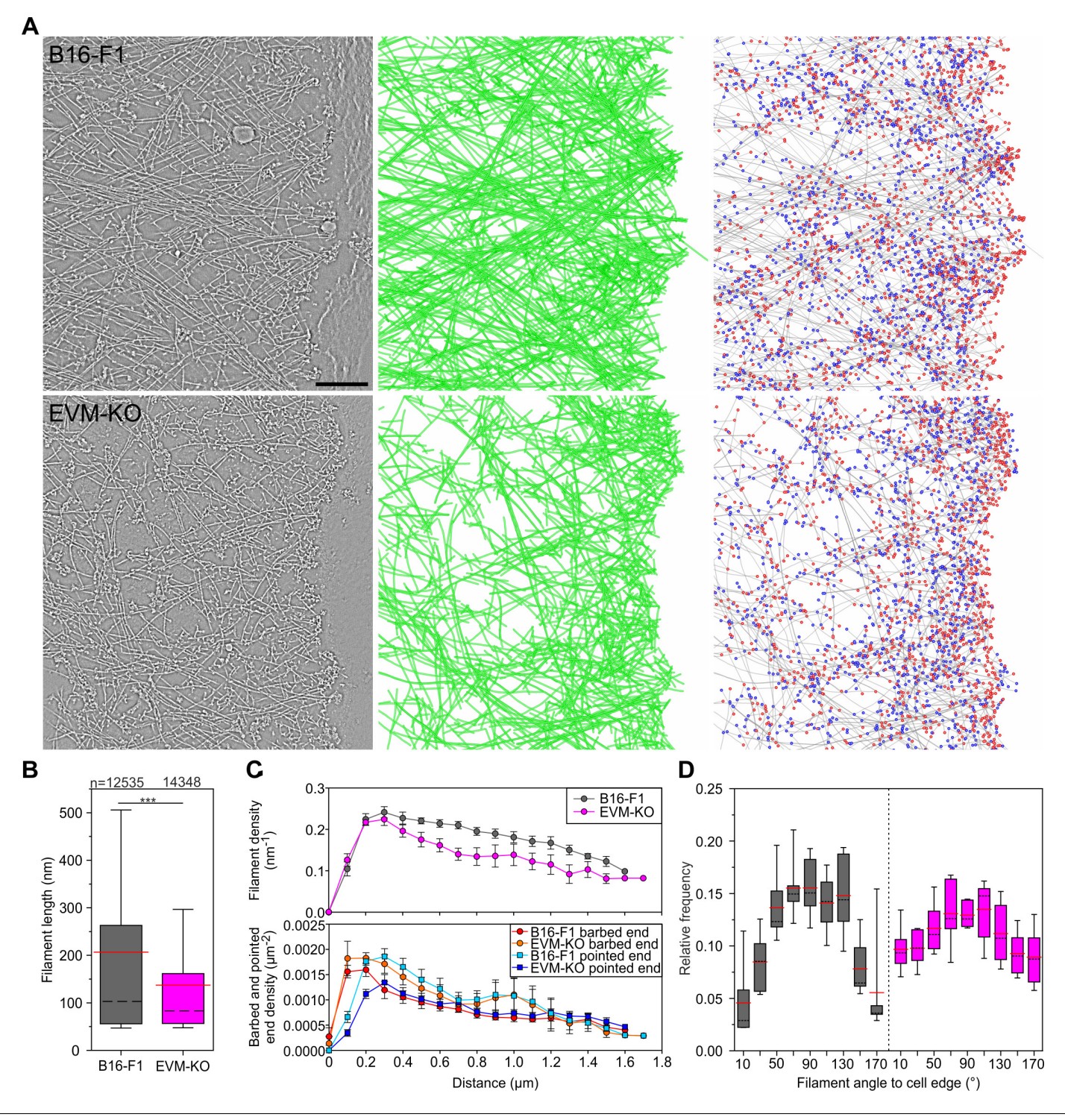

**Figure 6.** Electron tomography of ultrastructural changes in lamellipodial actin networks. (**A**) Transmission electron micrographs of representative wild-type (B16–F1) and EVM-KO cells showing distinct actin filament networks at the leading edge (left), and 2D projections of digital 3D tomograms showing either actin filament trajectories in green (middle), or barbed ends on grey filaments in red, and pointed ends in blue (right). Scale bar, 100 nm. (**B**) Quantification of filament length in wild-type and EVM-KO cells. ***p≤0.001 by Mann-Whitney U rank sum test. Whiskers indicate 10% and 90% confidence intervals. n indicates the number of filaments analyzed. Dashed black lines show median, red lines arithmetic mean. (**C**) Densities of filaments (*upper panel*), barbed and pointed ends (*lower panel*) in 106 nm-wide spatial bins throughout the lamellipodium. Error bars indicate SEM. 6 tomograms for each cell line were analyzed. (**D**) Histogram showing frequencies of filament angles to the leading edge (90° corresponding to filaments perpendicular to the leading edge). Dashed black lines are medians, and red lines arithmetic means.

*Figure 6 continued on next page*

*Figure 6 continued*

The online version of this article includes the following video and source data for figure 6:

**Source data 1.** Source data for details of lamellipodial actin networks including filament length, filament, barbed and pointed end densities and relative frequencies of filament angles *Figure 6*.

**Figure 6—video 1.** Loss of Ena/VASP deteriorates lamellipodium architecture in B16-F1 cells, related to *Figure 6*.

https://elifesciences.org/articles/55351#fig6video1

## Loss of Evl impairs cell spreading and FA formation

To assess Ena/VASP functions in adhesion, we first analyzed spreading of MV$^{D7}$, derived MVE-KO and reconstituted fibroblasts expressing EGFP-Evl on fibronectin by time-lapse imaging (*Figure 7A*, *Figure 7—video 1*). Notably, MVE-KO cells displayed a substantial spreading defect that was fully rescued by Evl re-expression (*Figure 7B–C*, *Figure 7—source data 1*). Then, we examined FA morphologies in MVE-KO fibroblasts expressing different Ena/VASP family members fused to EGFP. Untransfected MVE-KO cells formed noticeably smaller FAs on average than the same cells expressing distinct Ena/VASP family members, but the rescue with Evl appeared most effective (*Figure 7D*). To explore potential effects and differences of Ena/VASP proteins on FA formation more systematically, MV$^{D7}$, MVE-KO and reconstituted cell lines migrating on fibronectin were labeled for the FA-marker protein vinculin and assessed for various features (*Figure 7E*, *Figure 7—source data 1*). Images captured at identical settings were processed by a web-based FA analysis tool (*Berginski and Gomez, 2013*), allowing a global and unbiased assessment of multiple parameters. Consistent with previous work, untransfected MV$^{D7}$ double-mutant cells still contained prominent FAs (*Bear et al., 2000*). Notably however, vinculin intensity was clearly reduced in the triple mutant by almost 30% as compared to the parental MV$^{D7}$ cell line, and again fully rescued by Evl expression, while rescue of MVE-KO cells with VASP or Mena was less effective (*Figure 7F*, *Figure 7—source data 1*). Consistently, FA size in MVE-KO cells was reduced by 18% as compared to MV$^{D7}$ cells (*Figure 7G*, *Figure 7—source data 1*), despite the low abundance of Evl in this cell line (*Figure 2J*). Furthermore, expression of VASP or Mena in MVE-KO cells increased FA size only by 11% or 16%, respectively, while expression of Evl increased FA size by more than 24%. Accordingly, Evl expression was most efficient in rescuing the average number of FAs per cell as compared to VASP and Mena (*Figure 7H*, *Figure 7—source data 1*). Consistent with the removal of Evl in MV$^{D7}$ fibroblasts, B16-F1-derived EVM-KO cells also spread considerably slower than wild-type or reconstituted mutant cells expressing either EGFP-tagged VASP or Evl (*Figure 7—figure supplement 1A–B*). In these experiments, once more, Evl rescued more effectively than VASP (*Figure 7—figure supplement 1C*). Additionally, we also compared various adhesion parameters in EVM-KO and B16-F1 control cells and observed diminished vinculin intensity in FAs as well as reduced FA size and length (*Figure 7—figure supplement 1D–H*). To explore the more dominant role of Evl as for instance compared to VASP in adhesion in further detail, we performed FRAP experiments in EVM-KO cells of either EGFP-tagged Evl or VASP in focal adhesions (*Figure 7—figure supplement 2A–B*, *Figure 7—video 2*). Notably, Evl showed a moderate, but statistically significant increase in the amount of immobile fraction in focal adhesions as compared to VASP, potentially explaining its more robust accumulation in these structures as compared to EGFP-tagged VASP (*Figure 7—figure supplement 2B*, *Figure 7—video 2*). No differences, however, between these proteins were observed for their turnover rates, neither in focal adhesions nor lamellipodia (*Figure 7—figure supplement 2C–D*, *Figure 7—video 3*), suggesting that the observed, functional relevance of Evl in focal adhesions may indeed derive from its more stable association with specific focal adhesion components. Together, despite clearly overlapping functions of Ena/VASP proteins in various actin-dependent processes, Evl appeared more critical in the regulation of cell-substrate adhesions.

## Loss of Evl diminishes generation of traction forces

Since Evl proved most effective in rescuing parameters addressing FA properties, we finally explored the functional consequences of Evl loss of function and rescue in traction force development. To compare traction forces exerted onto a polyacrylamide substratum independently of cell size and geometry, MV$^{D7}$, MVE-KO cells, and reconstituted cells expressing EGFP-Evl were grown on top of crossbow-shaped micropatterns coated with fibronectin (*Vignaud et al., 2014*; *Figure 8A*).

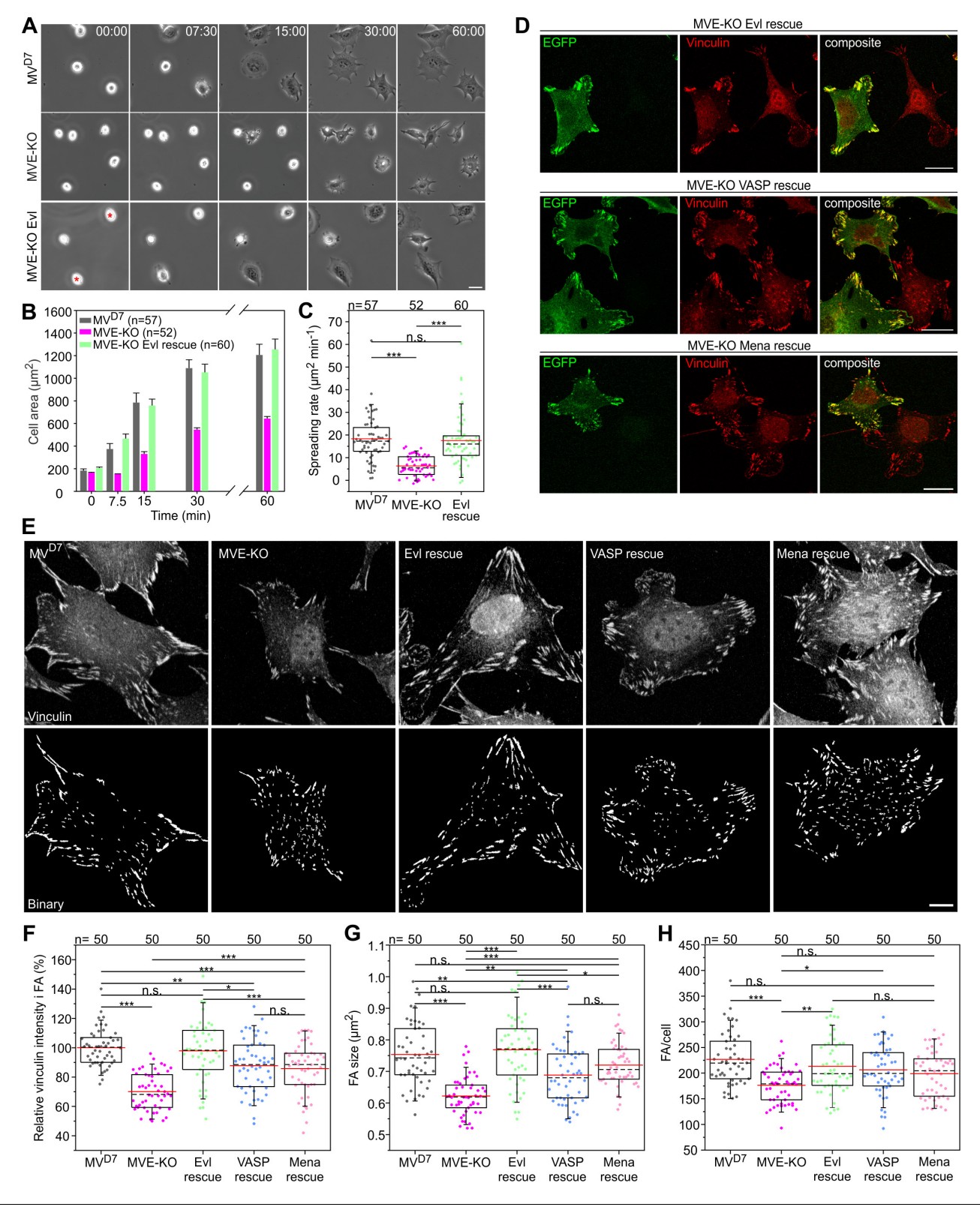

**Figure 7.** Inactivation of Evl in MV^D7 cells impairs FA formation. (**A**) Spreading of MV^D7, EVM-KO and reconstituted cells on fibronectin. EGFP-Evl expressing cells are marked with red asterisks. Time is in minutes. Bar, 25 μm. (**B**) Quantification of cell area over time. Data are means ± SEM. n, number of analyzed cells from five independent experiments. (**C**) Quantification of spreading rate. (**D**) Expression of EGFP-tagged Evl, -VASP or -Mena in MVE-KO cells promotes the formation of FA. Immunolabeling of EGFP and vinculin in cells seeded on fibronectin. Bars, 20 μm. (**E**) Representative

*Figure 7 continued on next page*

*Figure 7 continued*

micrographs of MV[D7], MVE-KO and reconstituted cells displaying vinculin staining before (upper panel) and after processing by Focal Adhesion Analysis Server (lower panel). Bar, 10 μm. (F) Quantification of vinculin intensities in FA. (G) Quantification of FA size. (H) Quantification of FA number per cell. (C and H) *p≤0.05, **p≤0.01, ***p≤0.001 and n.s., not significant by Kruskal-Wallis test and Dunn's Multiple Comparison test or by one-way ANOVA and Tukey Multiple Comparison test (F–G), respectively. n, number of analyzed cells from three independent experiments. Boxes in box plots indicate 50% (25–75%) and whiskers (5–95%) of all measurements, with dashed black lines depicting the medians, arithmetic means are highlighted in red.

The online version of this article includes the following video, source data, and figure supplement(s) for figure 7:

**Source data 1.** Source data for details of cell spreading including cell areas and spreading rates, and of FA parameters including relative vinculin intensities, FA sizes and numbers per cell *Figure 7*.
**Figure supplement 1.** Loss of Ena/VASP proteins affects cell spreading and FA formation in B16-F1 cells.
**Figure supplement 1—source data 1.** Source data for details of cell spreading including cell areas and spreading rates, and of FA parameters including relative vinculin intensities, FA sizes, length and widths *Figure 7—figure supplement 1*.
**Figure supplement 2.** The fraction of Evl stably associated with focal adhesions is increased as compared to VASP.
**Figure supplement 2—source data 1.** Source data for details of FRAP experiments including FA and lamellipodia *Figure 7—figure supplement 2*.
**Figure 7—video 1.** Loss of Evl in MV[D7] cells diminishes spreading on fibronectin, related to *Figure 7*.
https://elifesciences.org/articles/55351#fig7video1
**Figure 7—video 2.** FRAP of EGFP-tagged Evl *versus* -VASP in focal adhesions, related to *Figure 7—figure supplement 2*.
https://elifesciences.org/articles/55351#fig7video2
**Figure 7—video 3.** FRAP of EGFP-tagged VASP *versus* -Evl at the lamellipodium tip, related to *Figure 7—figure supplement 2*.
https://elifesciences.org/articles/55351#fig7video3

---

Interestingly, the contractile energy was significantly reduced in MVE-KO cells as compared to MV[D7] (*Figure 8B–C*, *Figure 8—source data 1*), and rescued upon stable expression of EGFP-tagged Evl (*Figure 8D–E*, *Figure 8—source data 1*). Moreover, the contractile energies of Evl-rescued MVE-KO and parental MV[D7] cells were highly similar to each other. Comparable results were obtained with unconfined cells, i.e. homogenous fibronectin coating on polyacrylamide substrata, and thus in the absence of size- and geometry-specific constraints (*Figure 8—figure supplement 1A–B*).

## Discussion

At variance to earlier studies (*Bear et al., 2002*; *Bear et al., 2000*; *Loureiro et al., 2002*), we show that Ena/VASP proteins positively regulate 2D cell migration of distinct mesenchymal cell types, since their consecutive loss in B16-F1 cells results in cumulative, reconstitutable motility phenotypes. This finding is not entirely surprising, given that Ena/VASP proteins localize to the leading edge and act as potent actin polymerases (*Breitsprecher et al., 2011a*; *Breitsprecher et al., 2008*; *Brühmann et al., 2017*; *Hansen and Mullins, 2010*), which are expected to generate actin polymer that drives motility by pushing against the membrane. Thus, despite increased directionality, the reduced net cell translocation rates observed in EVM-KOs as compared to wild-type occur because of reduced lamellipodial actin assembly and protrusion rates. Diminished migration was also observed in MV[D7] fibroblasts after genetic removal of Evl as well as in NIH 3T3 fibroblast after consecutive disruption of Mena and VASP, supporting the view that Ena/VASP operate in a mechanistically conserved fashion at least in adhesive, 2D migration, reconciling previous controversies (*Auerbuch et al., 2003*; *Geese et al., 2002*; *Lacayo et al., 2007*; *Loisel et al., 1999*; *Rottner et al., 1999*).

In line with markedly diminished 2D migration, EVM-KO cells display aberrant, narrow lamellipodia with reduced F-actin densities, shorter filaments and perturbed network geometry. But what are the precise consequences of loss of Ena/VASP activity in those lamellipodia still formed in their absence? Interestingly, lamellipodial accumulation of Arp2/3 complex, CP and cortactin is markedly enhanced upon Ena/VASP removal, but how can these changes be explained? During filament elongation, Ena/VASP proteins are thought to compete with CP for growing filament barbed ends (*Bear et al., 2002*; *Breitsprecher et al., 2011a*; *Breitsprecher et al., 2008*; *Hansen and Mullins, 2010*). One possibility therefore is that the absence of Ena/VASP may allow for an increased fraction of CP localizing to filament barbed ends in the lamellipodium tip. This effect might even be enhanced by the increase of barbed ends deduced from electron tomography data. An increased

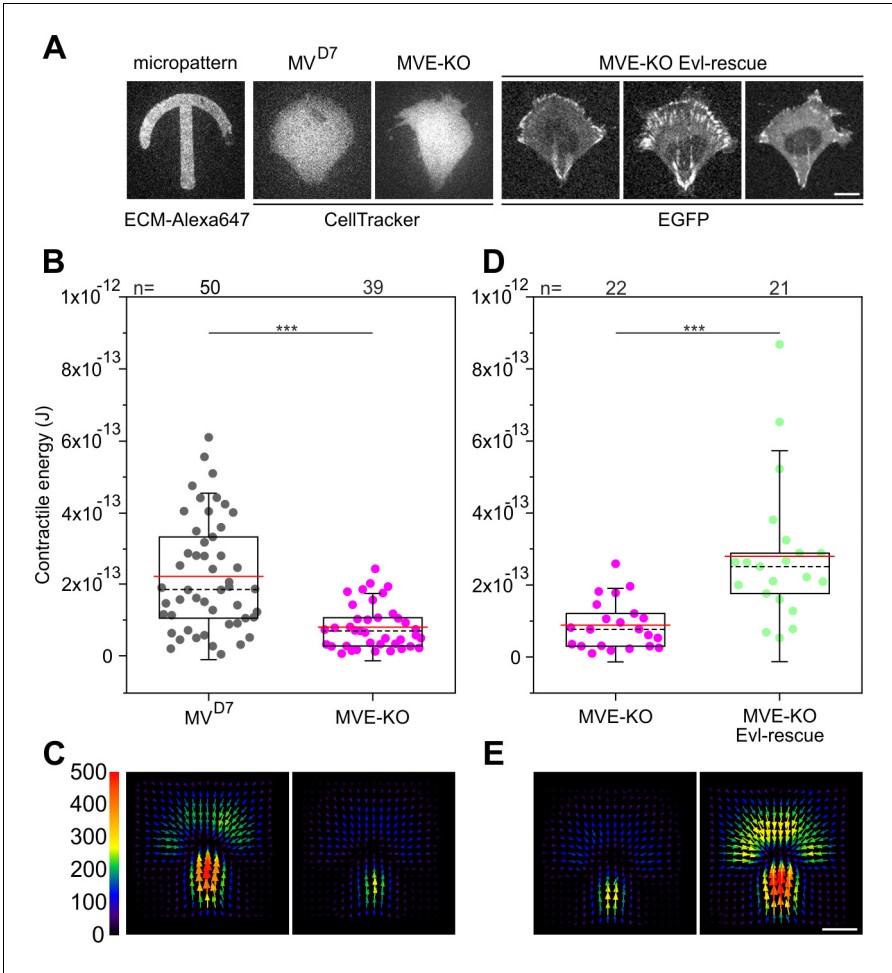

**Figure 8.** Confined MVE-KO cells exhibit diminished traction forces. (**A**) Representative micrographs of cells plated on crossbow-shaped polyacrylamide micropatterns coated with fibronectin. MV[D7] and MVE-KO were stained with CellTracker Green and rescued cells were stably expressing EGFP-Evl. Bar, 10 μm. (**B and D**) Quantifications of contractile energy of MV[D7], MVE-KO and Evl expressing cells plated on crossbow-shaped polyacrylamide micropatterns in independent experiments. ***p≤0.001 by Mann-Whitney U rank sum test. n, number of cells analyzed. Boxes in box plots indicate 50% (25–75%) and whiskers (5–95%) of all measurements, with dashed black lines depicting the medians, arithmetic means are highlighted in red. (**C and E**) Images depicting averaged traction force field representations of cells indicated. Bar, 10 μm. Force scale bar is in Pascal and arrows represent the local force magnitude and orientation.

The online version of this article includes the following source data and figure supplement(s) for figure 8:

**Source data 1.** Source data for details of contractile energies *Figure 8*.
**Figure supplement 1.** Inactivation of Evl in unconfined MV[D7] cells diminishes cell contractile energy.
**Figure supplement 1—source data 1.** Source data for details of contractile energies *Figure 8—figure supplement 1*.

amount of CP might then be sufficient to increase the activity of Arp2/3 complex, as CP has also been reported to operate as co-factor of Arp2/3-dependent actin assembly (*Akin and Mullins, 2008*). In line with this view, overexpression of VASP in B16-F1 cells was previously found to increase F-actin, but decrease Arp2/3 complex intensities in lamellipodia (*Dimchev et al., 2017*). However, the latter finding could also speak for an alternative explanation of our observations: In fact, VASP-mediated suppression of Arp2/3 complex incorporation into lamellipodial networks would also be consistent with the previously observed increase of branch spacing of filaments in Arp2/3-dependent actin tails, as effected by Ena/VASP family members in vitro (*Pernier et al., 2016*; *Samarin et al., 2003*). Thus, upon loss of Ena/VASP, filaments may not only become shorter, but Arp2/3 complex

intensities simply increase due to enhanced branching activity relative to growing actin networks, which would also be consistent with the shorter and more highly branched filaments previously observed by platinum replica EM upon mitochondrial Ena/VASP sequestration (*Bear et al., 2002*). In this alternative possibility, increased Arp2/3 densities might in turn indirectly contribute to the increase of its co-regulator CP (*Akin and Mullins, 2008*). The latter view is supported by the observation that acute, experimental sequestration of Arp2/3 complex, achieved by microinjection of recombinant WCA, coincides with removal of both Arp2/3 complex and CP from lamellipodia, suggesting that capping protein accumulation in the lamellipodium does indeed depend on the presence of Arp2/3 complex (*Koestler et al., 2013*). The increased accumulation of cortactin in mutant lamellipodia is consistent with the general coincidence of its subcellular positioning with Arp2/3 complex (*Schnoor et al., 2018*). Future studies perhaps involving cryo-EM tomography will have to define which one of these possibilities – if not both - will actually hold true.

This notwithstanding, EVM-KO cells are still capable of lamellipodia formation and migration, albeit in a significantly compromised fashion, raising the question of alternative actin polymerases partly taking over. Previous studies have implicated formins in lamellipodia protrusion in B16-F1, including mDia2 (*Yang et al., 2007*) and FMNL 2 and −3 (*Kage et al., 2017*). Indeed, FMNL2/3 removal caused defects partly reminiscent of what's seen here, at least concerning impact on lamellipodial protrusion, actin filament densities or microspike formation (see below), although phenotypes seen upon Ena/VASP deficiency were more severe. Thus, the extent of potential redundancy between all these factors will only be revealed by systematic, combinatorial loss of function studies and side-by-side, phenotypic comparison (*Young et al., 2018*). Derived conclusions will also have to consider the exciting, recently discovered 'distributive polymerase' activity ascribed to WAVE (*Bieling et al., 2018*).

As yet, authors of even highly relevant literature have decided not to clearly distinguish between filopodia and microspikes to facilitate analyses of finger-like protrusions (*Svitkina et al., 2003*; *Vignjevic et al., 2006*). However, the virtual absence of microspikes, but not of filopodia, in EVM-KO cells combined with the inability of established filopodia inducers such as mDia2, FMNL3, myosin X and Rif to rescue microspikes clearly establishes, perhaps for the first time, fundamental differences between these molecular entities. The mechanisms underlying filopodia assembly are still under debate: In the convergent elongation model, filopodia form by coalescence of lamellipodial filaments (*Svitkina et al., 2003*), whereas the de novo nucleation model proposes filopodia to arise independently of lamellipodial actin networks, as effected by formin-mediated filament nucleation and elongation (*Faix et al., 2009*; *Mellor, 2010*). The findings provided here strongly suggest that in contrast to filopodia, microspikes form exclusively by convergent elongation, as previously suggested (*Svitkina et al., 2003*), whereas filopodia may emerge by multiple, distinct mechanisms (*Young et al., 2015*), opening a door for resolving a long-standing controversy in the field.

Ena/VASP proteins are well established to accumulate in FAs of adherent cells, and known to reside together with vinculin in the force-transduction layer in-between talin and ends of stress fiber filaments (*Kanchanawong et al., 2010*). The binding of vinculin to F-actin is critical to its role in cell-matrix adhesion since disruptions of the vinculin-F-actin interaction affect cell morphology, cell motility and adhesion (*Ezzell et al., 1997*). The underlying cause of these defects is apparently the impaired ability to generate and transduce forces (*Mierke et al., 2008*). Of note, besides their actin polymerase activities, Ena/VASP proteins are ideally suited for the capture and tethering of dynamic filament barbed ends (*Bear et al., 2002*; *Breitsprecher et al., 2011a*; *Breitsprecher et al., 2008*). Moreover, since vinculin and VASP interact with each other (*Brindle et al., 1996*), vinculin-Ena/VASP complexes might synergize in the capture of filament barbed ends, to regulate force generation and transduction in FAs. However, the presence of multiple Ena/VASP orthologues and analyses of only partially-depleted cells hitherto prevented careful functional assessment of their precise contributions to integrin-dependent adhesion. Here, we show that lack of Ena/VASP proteins consistently impairs integrin-dependent spreading and changes in FA patterns, and that from the three Ena/VASP family members, the rescue with Evl is most effective concerning relevant adhesion parameters. The particularly important role of Evl in adhesion is strengthened by diminished traction forces upon its disruption, and further supported by our FRAP data, implying that Evl is more stably associated with other focal adhesion components as compared to VASP. All this clearly illustrates the positive regulatory function of Ena/VASP family members in adhesion. The specialized role of Evl in

integrin-dependent adhesion is consistent with very recent work showing its specific requirement for FA maturation, cell-matrix adhesion and mechanosensing (*Puleo et al., 2019*).

Finally, we have not only sought to interpret all our phenotypes in the context of various combinations of gene deletions employing up to three distinct mesenchymal cell types, but also assessed their expression levels in different cell types relative to each other, which certainly adds an additional layer of complexity. Two major points being more systematically followed in the future are worth mentioning here. First, Mena is by far the most abundant protein present in both melanoma and fibroblast cell lines, but in particular considering this fact, phenotypes generated by its elimination are comparably modest. In spite of its abundance, we are tempted to speculate therefore that it simply does not sufficiently find its relevant, specific interactors that might promote its impact in other structures such as cell-cell contacts (*Scott et al., 2006*) or tissues like the nervous system (*Kwiatkowski et al., 2007*). Finally, given their comparably lower abundance, VASP and Evl are instead displaying a surprisingly significant relevance in lamellipodia protrusion and adhesion formation, respectively. Notwithstanding this, VASP is still expressed at much higher levels compared to Evl, which by itself could explain its more prominent role in cell migration. Clearly, much remains to be learned concerning both redundancies and specific functions that can be assigned to individual Ena/VASP family members, in particular in various cell types and tissues throughout the mammalian body, but the experimental approaches and tools developed here constitute first steps in this direction.

## Materials and methods

Please refer to *Supplementary file 1*: Key resources table for details of resources used and created for this study.

### Constructs

Constructs cDNAs of *Vasp* encoding murine VASP (isoform 1), *Enah* encoding Mena (isoform 2), and *Evl* encoding Evl (isoform 2), respectively, were amplified by PCR from a NIH 3T3 cDNA library and ligated into suitable sites of pEGFP-C1 (Clontech, Palo Alto, CA). *Vasp* cDNA was additionally inserted into the BglII and SalI sites of pmCherry (Addgene ID: 632524). For generation of stably transfected cell lines, the 1.3 kb Puro cassette was amplified by PCR using pPur (Clontech) as template and inserted into the Asp718 and BamHI sites of pEGFP-C1 to yield pEGFP-C1 Puro. Subsequently, the *Evl* cDNA fragment was inserted into the BspEI and XhoI sites of pEGFP-C1 Puro. For generation of constitutively active murine mDia2 (Drf3) lacking the C-terminal regulatory DAD domain (1–1.036), the corresponding cDNA fragment was amplified from a NIH 3T3 cDNA library and inserted into the BspEI and Asp718 sites of pEGFP-C1. Plasmids pEGFP-C1 Lifeact-EGFP (*Riedl et al., 2008*), pEGFP-Rif-L77 (*Aspenström et al., 2004*), pEGFP-MyoX (*Berg and Cheney, 2002*), EGFP-FMNL2-E272 and EGFP-FMNL3-E275 (*Kage et al., 2017*) and pEGFP-fascin *Adams and Schwartz, 2000* have been described. pEGFP-β-Actin was from Clontech. For generation of recombinant antigens, respective sequences encoding for murine Evl (1-414), VASP (1-375), Mena (1-241), and WAVE2-WCA (419-497) were amplified from plasmid DNA and inserted into BamHI/SalI sites of pGEX-6P-1 (GE Healthcare, Munich, Germany). Plasmids pGEX-6P-2-cortactin (*Lai et al., 2009*) and pGEX-6P-1-fascin *Breitsprecher et al., 2011b* have been described. Fidelity of generated plasmids was confirmed by sequencing.

### Cell culture and transfection

B16-F1 mouse melanoma cells (CRL-6323) and mouse embryonic NIH 3T3 fibroblasts (CRL-1658) were purchased from ATCC. The mouse embryonic fibroblast cell line MV$^{D7}$, which expresses a temperature-sensitive version of large T antigen, was derived from Mena/VASP-deficient mice (*Bear et al., 2000*). For this study, MV$^{D7}$ cells were additionally immortalized with a retrovirus transducing a wild-type variant of SV40-large T (LT) antigen using standard procedures, to allow cultivation at 37°C. Expression of Evl in original and MV$^{D7}$ cells immortalized with wild-type, SV40-LT antigen was unchanged, as assessed by immunoblotting. B16-F1, NIH 3T3 and wildtype SV40-LT-immortalized MV$^{D7}$ cells were cultured at 37°C and 5% $CO_2$ in high-glucose DMEM culture medium (Lonza, Cologne, Germany) supplemented with 10% FBS (Biowest), 2 mM UltraGlutamine (Lonza) and 1% Penicillin-Streptomycin (Biowest). B16-F1 cells were transfected with 1 µg plasmid DNA and

MV^D7 and NIH 3T3 cells with 3 μg plasmid DNA, respectively using JetPRIME transfection reagent (PolyPlus, Illkirch, France) in 35 mm diameter wells (Sarstedt, Nümbrecht, Germany) according to the manufacturer's protocol. Used cell lines are routinely authenticated following common guide lines by local authorities. Absence of mycoplasma in cell lines was routinely checked by the VenorGeM Mycoplasma Detection Kit (Sigma, St. Louis, MO).

## Genome editing by CRISPR/Cas9

DNA target sequences were pasted into a CRISPR/Cas9 design tool (http://crispr.mit.edu/) to generate sgRNAs of 20 nucleotides with high-efficiency scores. In case of *Evl*, the targeting sequence 5'-GATCGGTACCCACTTCTTAC-3' was used to cover all possible splice variants of the gene. Accordingly, genome editing of *Vasp* was performed with 5'-GTAGATCTGGACGCGGCTGA-3', and for disruption of the *Enah* gene, the sgRNA 5'-AAGGGAGCACGTGGAGCGGC-3' was used. Respective sequences were ligated into expression plasmid pSpCas9(BB)-2A-Puro(PX459)V2.0 (Addgene plasmid ID: 62988) using BbsI (*Ran et al., 2013*). Validation of CRISPR construct sequences was performed using a 5'-GGACTATCATATGCTTACCG-3' sequencing primer. 24 hr after transfection with the CRISPR constructs, cells were selected in culture medium containing puromycin (B16-F1: 2 μg/mL, MV^D7: 4 μg/mL and NIH 3T3: 5 μg/mL) for 4 days, and then cultivated for 24 hr in the absence of puromycin. For isolation of clonal knockout cell lines, single cells were seeded by visual inspection into 96-well microtiter plates and expanded in pre-conditioned culture medium containing 20% Bri-Clone supplement (NICB, Dublin City University, Ireland). Clones were analyzed by the TIDE sequence trace decomposition web tool (https://tide.deskgen.com/; *Brinkman et al., 2014*) and immunoblotting using specific antibodies, and subsequently verified by sequencing (*Supplementary file 1*). For the latter, respective DNA fragments of about 500 bp encompassing the target sites were first amplified from genomic DNA using the Evl primer combination 5'-AAGCCATGAGTCTCCCAAGC-3' and 5'-GTCTCACGCTTTGGCTCTCA-3', the VASP primer combination 5'-GTGTGGCCTGCCTATCTGTT-3' and 5'-CAGAGGGACAGAGGGACAGA-3', and the Mena primer pair 5'-TCAGGCAACTGCAAGAACAG-3' and 5'-CATCTCGGCTGTAGGAGGTG-3'. Subsequently, amplified fragments were inserted into pJet1.2 vector (Thermo Fisher Scientific, Carlsbad, CA) and used for transformation of *E. coli* host DH5α (Thermo Fisher Scientific). At least 30 sequences were analyzed for each knockout clone.

## Generation of recombinant proteins and antibodies

For expression of recombinant proteins, the *E. coli* strain Rossetta 2 (Novagen) was used. Expression of GST-tagged fusion proteins was induced using 1 mM isopropyl-ß-D-thiogalactoside (IPTG) (Carl Roth, Karlsruhe, Germany) at 21°C for 14 hr. Purification of respective proteins from bacterial extracts was performed by affinity chromatography using Protino glutathione-conjugated agarose 4B (Macherey-Nagel, Düren, Germany). Except for GST-WAVE2-WCA, the GST-tag was removed by proteolytic cleavage with PreScission Protease (GE Healthcare), followed by a final polishing step of the proteins by size-exclusion chromatography on an Äkta Purifier System using either HiLoad 26/600 Superdex 200 or HiLoad 26/75 Superdex columns (GE Healthcare). Purified proteins were dialyzed against immunization buffer (150 mM NaCl, 1 mM dithiothreitol (DTT), and 20 mM Tris/HCl pH 7.4) and stored in aliquots at −20°C. Polyclonal antibodies against Evl, VASP, Mena, cortactin and WAVE2 were raised by immunizing New Zealand white rabbits with respective recombinant proteins following standard procedures. Evl, Mena and cortactin polyclonal antibodies were subsequently purified by affinity chromatography using antigens conjugated to sepharose. GST antibodies in the anti GST-WAVE2-WCA serum were absorbed by GST-affinity chromatography. Monoclonal antibodies against fascin were produced upon injecting recombinant fascin mixed with CpG-DNA as adjuvant (*Magic Mouse*, Creative, NY, USA) into 7 week-old, female mice (Charles River, USA) using standard hybridoma technology and antibody screening procedures in the PROCOMPAS graduate programme-funded monoclonal antibody facility (Braunschweig Integrated Centre of Systems Biology - BRICS, Technische Universität Braunschweig). Hybridoma clone 5E2 was selected and kindly provided by Sabine Buchmeier and Prof. Dr. em. Brigitte Jockusch (TU Braunschweig).

## Antibodies used

Immunoblotting was performed according to standard protocols using rabbit polyclonal antibodies directed against Evl (1:1000 dilution), VASP (1:1000 dilution), Mena (1:1000 dilution), WAVE2 (1:1000 dilution), GFP (1:2000) (*Faix et al., 2001*) or mouse monoclonal antibody against glyceralde-hyde-3-phosphate dehydrogenase (GAPDH) (1:1000; #CB1001-500UG, Merck (Darmstadt, Germany)) and anti-fascin antibody 5E2 (undiluted hybridoma supernatant). Primary antibodies in immunoblots were visualized using phosphatase-coupled anti-mouse (1:1000 dilution; #115-055-62, Dianova (Hamburg, Germany)) or anti-rabbit antibodies (1:1000 dilution; #115-055-144, Dianova). For immunofluorescence, the following primary antibodies were used: rabbit anti-cortactin antibodies (1:1000 dilution), mouse monoclonal anti-fascin antibody 5E2 (undiluted hybridoma supernatant), mouse monoclonal anti-ArpC5A (p16-ArcA) from hybridoma culture clone 323H3 (*Olazabal et al., 2002*), mouse monoclonal anti-capping protein $\alpha1/\alpha2$ subunits mAb B5 12.3 (hybridoma supernatant; 1:4 dilution; Developmental Hybridoma Bank (deposited by J. Cooper), University of Iowa, Iowa City, IA,) and mouse monoclonal anti-vinculin antibody (1:1000 dilution, #V9131, clone hVIN-1, Sigma). Primary antibodies were visualized in immunohistochemistry with polyclonal Alexa-555-conjugated goat-anti-rabbit (1:1000 dilution; #A21429, Invitrogen (Carlsbad, CA)) or goat-anti-mouse (1:1000 dilution, #A32727, Invitrogen) and Alexa-488-conjugated goat-anti-rabbit (1:1000 dilution; #A-11034, Invitrogen) or goat-anti-mouse antibodies (1:1000 dilution; #A-11029, Invitrogen). To enhance EGFP signals, Alexa488-conjugated nanobodies from Chromotek (Chromotek, (Planegg-Martinsried, Germany)) (1:200 dilution; #gba488) were used. Atto550-phalloidin (1:250 dilution, #AD 550–82, Atto-Tec (Siegen, Germany)) was used for visualization of F-actin.

## Immunoblotting

For preparation of total cell lysates, cells were cultured to confluency and trypsinized. Cell pellets were washed twice with cold PBS and lysed with cold RIPA buffer (150 mM NaCl, 1.0% Triton X-100, 0.5% sodium deoxycholate, 0.1% sodium dodecyl sulfate (SDS), 50 mM Tris, pH 8.0) supplemented 5 mM benzamidine (Carl Roth), 0.1 mM AEBSF (AppliChem, Darmstadt, Germany) and Benzonase (1:1000, Merck) for 1 hr at 4°C on a wheel rotator. Cell lysates were subsequently homogenized by passing the lysate 10 times through a syringe cannula (Braun). Protein contents of total cell lysates were determined by Pierce BCA assay (Thermo Fisher Scientific) using a Synergy 4 fluorescence microplate reader (Biotek, Bad Friedrichshall, Germany) according to manufacturer's protocol. 50 µg (B16-F1) or 100 µg (MEF) of total proteins per lane were subjected to SDS-PAGE, and transferred by semi-dry blotting onto nitrocellulose membranes (Hypermol, Hannover, Germany). Blotting membranes were then blocked with NCP buffer (10 mM Tris/HCL, 150 mM NaCl, 0.05% Tween-20, 0.02% $NaN_3$, pH 8.0,) containing 4% bovine serum albumin (BSA) for 1 hr and incubated with primary antibodies overnight in the same buffer. After extensive washing of membranes and incubation with secondary, phosphatase-conjugated antibodies for at least 2 hr, blots were developed with 20 mg/mL of 5-brom-4-chlor-3-indolylphosphat-p-toluidin (BCIP) in $NaHCO_3$, pH 10.0.

## Quantification of cellular Ena/VASP protein concentrations

Total cellular concentrations of Ena/VASP family members in used cell lines were quantified from immunoblots using orthologue-specific antibodies by titrating total cell lysates corresponding to a defined number of B16-F1, NIH 3T3 and MV$^{D7}$ cells with dilution series containing defined amounts of recombinant proteins loaded on the same gels. Band intensities were quantified by ImageJ software. The cell volume for each cell type was calculated from phase-contrast images of freshly trypsinized, non-adherent cells (>50 for each cell line). The volume of the nucleus was subtracted from the cell volume to obtain the volume of the cytoplasm.

## Live cell imaging

Time-lapse imaging of cells was performed using an Olympus XI-81 inverted microscope (Olympus, Hamburg, Germany) driven by Metamorph software (Molecular Devices, San Jose, CA) and equipped with objectives specified below and a CoolSnap EZ camera (Photometrics, Tucson, AZ). Cells were seeded onto 35 mm glass bottom dishes (Ibidi, Planegg-Martinsried, Germany) coated with either 25 µg/mL laminin (Sigma) in case of B16-F1 cells and derived clones, or with 10 µg/mL fibronectin (Roche, Penzberg, Germany) in case of MV$^{D7}$, NIH 3T3 and their derivatives, and

maintained in imaging medium composed of F-12 Ham Nutrient Mixture with 25 mM HEPES (Sigma), the latter to compensate for the lack of $CO_2$, and supplemented with 10% FBS (Biowest, Nuaillé, France), 1% Penicillin-Streptomycin (Biowest), 2 mM stable L-glutamine (Biowest), and 2.7 g/ L D-glucose (Carl Roth) in an Ibidi Heating System at 37°C. For random motility assays, B16-F1cells were seeded at low density onto the dishes and allowed to adhere for 3 hr. Subsequently, the medium was exchanged with imaging medium, the chamber mounted into a heating system, and cells recorded by time-lapse phase-contrast imaging at 60 s intervals for 3 hr using an Uplan FL N 4x/0.13NA objective (Olympus). NIH 3T3, MV$^{D7}$ and derived cells were allowed to settle for 6 hr after seeding and imaged at 10 min intervals for 10 hr using an UPlan FL N 10x/0.30NA objective (Olympus). Single cell tracking was performed in ImageJ by MTrackJ. Analyses of cell speed and cell trajectories, turning angles, and mean square displacements were performed in Excel (Microsoft, Redmond, WA) using a customized macro (*Litschko et al., 2018*). Cells that contacted each other or divided were excluded from analyses. Directionality index ratio was determined by dividing the shortest distance between starting and end points (d) by the actual trajectories (D). For wound healing assays, MV$^{D7}$ cells were seeded onto uncoated dishes and expanded to confluency. Subsequently, the monolayer was scratched with a 200 µL pipette tip, the cells washed three times with warm imaging medium, and recorded by phase-contrast time-lapse imaging at 10 min intervals for 20 hr using an UPlan FL N 10x/0.30NA objective (Olympus). Wound closure rates were determined in ImageJ by measuring decrease of scratch area over time. Lamellipodial protrusions were recorded at 5 s intervals for 10 min using an UPlan FL 40x/0.75NA objective (Olympus). Protrusion rates of advancing lamellipodia assessed over a time period of 2.5 min were quantified by first generating kymographs from time-lapse movies of the cell periphery using ImageJ, followed by slope determination from these kymographs. Lamellipodial persistence of randomly migrating B16-F1 wild-type and derived cells on laminin was determined by phase-contrast time-lapse microscopy using an UPlan FL N 10x/0.30NA objective and a frame rate of 1 frame per minute. Lamellipodial persistence was defined as time in min from initiation till collapse of the lamellipodium. Spreading of wild-type B16-F1 cells and derived clones was monitored at 30 s intervals by time-lapse phase-contrast imaging for 30 min using an UPlan FL N 10x/0.30NA objective immediately after seeding, and spreading of MV$^{D7}$ and derived cell lines were imaged at 30 s intervals for 1 hr using an Olympus LUCPlan FL N 20x/0.45NA objective. Quantification of cell spreading was executed with ImageJ from time-lapse movies by measuring increase of cell area over time. For epifluorescence imaging of fluorescently-labelled cells, B16-F1, EVM-KO and reconstituted EVM-KO cells expressing mCherry-VASP were transfected with either pEGFP-Lifeact or pEGFP-fascin. 24 hr post transfection, cells were seeded into imaging medium, and after 3 hr, migrating cells recorded by time-lapse imaging at 10 s intervals for 10 min using an Olympus 40x/0.75NA Uplan FL objective. In motility and spreading assays, reconstituted cells expressing EGFP-tagged VASP, Evl or Mena were identified by epifluorescence imaging.

The circular invasion assay was performed as described (*Yu and Machesky, 2012*), but with slight modifications. Specifically, B16-F1 wild-type and EVM KO (#23.7.66) cells were seeded into a dish termed Culture-Insert 4 Well in µ-Dish 35 mm, high (ThermoFisher Scientific, Ibidi, 80466), and allowed to reach confluence. The silicone insert was removed and cells were overlaid with 350 µl of Matrigel (Corning, Corning, NY) (4.5 mg/mL), diluted in DMEM, which was allowed to polymerize for 30 min at RT. Thereafter, microscopy medium [Nutrient Mixture F-12 Ham, supplemented with 10% FBS (Gibco, Thermo Fisher Scientific), 2 mM glutamine (Thermo Fisher Scientific) and penicillin (50 Units/mL)/streptomycin (50 µg/mL) (Thermo Fisher Scientific)] was added on top of the polymerized Matrigel. Subsequently, cells were subjected to live cell imaging for 40 hr, followed by determination of average invasion into the Matrigel.

## Immunofluorescence

If not indicated otherwise for immunofluorescence labelling, cells were fixed in pre-warmed, 4% PFA in PBS, pH 7.3 for 20 min, subsequently washed three times with PBS supplemented with 100 mM glycine, permeabilized with 0.1% Triton X-100 in PBS for 3 min and blocked in PBG (PBS, 0.045% cold fish gelatin (Sigma), and 0.5% BSA). For immunolabeling with fascin, cells were fixed in −20°C cold methanol. For immunolabeling of capping protein, the cells were fixed with 3% glyoxal at pH 5.0 as described (*Richter et al., 2018*) and then treated as above. To reduce cytoplasmic background in vinculin stainings, cells were fixed for 1 min with 2% PFA in PBS containing 0.3% Triton

X-100 and then postfixed with 4% PFA in PBS for 20 min as described (*Kage et al., 2017*). Primary antibodies were incubated overnight, followed by extensive washing of the specimens with PBG and incubation with respective secondary antibodies for at least 2 hr. GFP signals were enhanced with Alexa488-conjugated nanobodies. F-actin was visualized with Atto550 phalloidin. Imaging of fixed cells was performed with an Olympus XI-81 inverted microscope equipped with an UPlan FI 100x/1.30NA oil immersion objective or an LSM510 Meta confocal microscope (Carl Zeiss, Jena, Germany) equipped with a Plan-Neofluar 63x/1.3NA oil immersion objective using 488 nm and 543 nm laser lines.

Fluorescence intensities of phalloidin or lamellipodial proteins were measured from still images captured at identical settings using ImageJ software after background subtraction. Mean pixel intensities in lamellipodial regions of interest are shown as whiskers-box plots including all data points.

## Induction of filopodia in absence of lamellipodia

B16-F1 wild-type and EVM-KO cells were seeded onto low laminin (1 μg/mL) for 1 hr, a concentration lower than the threshold for inducing prominent lamellipodia in this cell type and conditions. Then, the cells were incubated for 2 hr with the Arp2/3 complex inhibitor CK666 (200 μM) (Sigma) to completely abolish lamellipodia and trigger filopodia formation. Subsequently, cells were fixed and stained with phalloidin for the actin cytoskeleton followed by confocal imaging.

## Fluorescence recovery after photobleaching (FRAP) and determination of actin assembly rates in lamellipodia

FRAP experiments were performed using a 100x/1.4NA Plan apochromatic oil immersion objective on an inverted Axio Observer microscope equipped with a DG4 light source (Sutter Instrument) for epifluorescence illumination and a Coolsnap-HQ2 camera (Photometrics) driven by VisiView software (Visitron Systems, Puchheim, Germany). In addition, bleaching was performed using a 405 nm diode laser controlled by the so called 2D-VisiFRAP Realtime Scanner (Visitron Systems, Puchheim, Germany).

For analyzing dwell times of VASP and Evl in lamellipodia and focal adhesions, EGFP-tagged VASP and -Evl were transfected into EVM KO cells, respectively. Movies were acquired at a rate of 2 s per frame. Photobleaching and analysis were essentially performed as previously described (*Lai et al., 2008*; *Steffen et al., 2013*), manually triggered using a 405 nm diode laser at 100–120 mW output power. Upon bleaching of rectangular regions covering focal adhesions or subfractions of lamellipodia and recording of recovery of fluorescence intensity values in bleached regions over time, analyses were initiated by subtracting extracellular regions (background). Correction for acquisition photobleaching was subsequently performed by normalizing data to average intensities over time obtained from cytosolic regions. Intensities were then normalized to the mean intensity before photobleaching (1) and directly after photobleaching (0). Fluorescence recovery in lamellipodia was modelled using one phase association equation [$Y(x)=a*(1-\exp(-b*x))$] assuming one dwell time (*Lai et al., 2008*), and recovery in focal adhesions was modelled using two phase association equation [$Y(x)=a*(1-\exp(-bx))+c*(1-\exp(-dx))$] assuming two dwell times as described (*Steffen et al., 2013*). To estimate mobile fractions for each individual experiment employed for statistical analysis, the last three values of respective curves were averaged.

Quantification of lamellipodial actin polymerization rates were performed in B16-F1 and EMV-KO cells transiently expressing EGFP-ß-actin, using FRAP in protruding lamellipodial regions followed by determination of the distance of fluorescence recovery from the lamellipodium tip over time, as described (*Dimchev and Rottner, 2018*). Bleaching of EGFP-ß-actin was performed essentially with aforementioned system, but with the diode laser at settings as follows: 65 mW laser power, 10-pixel laser beam diameter, 1 ms bleach dwell time/pixel, again manually triggered in selected regions during image acquisition. Lamellipodial EGFP-ß-actin FRAP movies were routinely recorded using exposures time of 500 ms and a frame rate of 1.5 s.

## Electron tomography

B16-F1 and EMV-KO cells were grown on Formvar-coated copper-palladium grids coated with 25 μg/mL laminin (Sigma) in laminin coating buffer (150 mM NaCl, 50 mM Tris, pH 7.5). The cells were simultaneously extracted and fixed for 1 min with 0.5% Triton X-100 (Fluka, Buchs, Switzerland) and

0.25% glutaraldehyde (Agar Scientific, Stansted, United Kingdom) in cytoskeleton buffer (10 mM MES buffer, 150 mM NaCl, 5 mM EGTA, 5 mM glucose and 5 mM MgCl$_2$, pH 6.1), followed by a second fixation step for 15 min in cytoskeleton buffer (pH 7.0) containing 2% glutaraldehyde. The coverslips were incubated for additional 4–12 hr in cytoskeleton buffer containing 2% glutaraldehyde at 4° C, and were then stained for electron microscopy. Negative staining was performed in mixtures of 4–6% sodium silicotungstate (Agar Scientific) at pH 7.0, containing 10 nm gold colloid saturated with BSA, and diluted 1:10 from a gold stock (*Urban et al., 2010*). Double tilt series of negatively stained cytoskeletons were acquired on a FEI Tecnai G20 transmission electron microscope (Tecnai, Hillsboro, OR) operated at 200 kV driven by SerialEM 3.x (*Mastronarde, 2005*) equipped with an Eagle 4 k HS CCD camera (Gatan, Pleasanton, CA). Double axis tilt series were acquired with typical tilt angles from −65° to +65° and 1° increments following the Saxton scheme at a primary on-screen magnification of 25,000x. Actin filaments were tracked automatically using a Matlab-based tracking algorithm (*Mueller et al., 2017*; *Winkler et al., 2012*).

## Focal adhesion analysis

Focal adhesions in B16-F1, MV$^{D7}$ and derived cells were analyzed from confocal images captured at identical settings. Images of cells labeled for vinculin were first processed in ImageJ by background subtraction using a rolling ball radius of 40 pixels, exclusion of nuclear regions due to high background staining, and by contouring of cell perimeters. Preprocessed images were then analyzed by the Focal Adhesion Analysis Server (*Berginski and Gomez, 2013*). The parameters were set for MV$^{D7}$ and derived cells to imaging frequency 0 min, detection threshold 3.5 and minimal adhesion size of 10 pixels. For B16-F1 and EVM-KO cells, an imaging frequency of 0 min, a detection threshold of 2 and a minimal adhesion size of 5 pixels were used. Obtained values for FA intensity and size as well as number of FAs/cell were subsequently analyzed in Excel.

## Traction force analysis

Patterned polyacrylamide hydrogels were fabricated according to the Mask method (*Vignaud et al., 2014*). A quartz photomask was first cleaned through oxygen plasma (AST product, 300W) for 3.5 min at 200 W. Areas containing crossbow patterns were then incubated with 0.1 mg/mL Poly(L-lysine)-graft-poly(ethylene glycol) (PLL-g-PEG) (JenKem Technology, Plano, TX) in 10 mM HEPES, pH 7.4 for 30 min. After a rapid de-wetting step, PLL-PEG was burned using deep-UV exposure for 6 min. Patterns on the mask were then incubated with a mix of 10 µg/mL fibronectin (Sigma) and 10 µg/mL fibrinogen Alexa Fluor 647 conjugate (Invitrogen) in 100 mM sodium bicarbonate buffer pH, 8.4 for 30 min. A mix of 8% acrylamide (Sigma) and 0.264% bis-acrylamide solution (Sigma) was degassed for 30 min, mixed with 0.2 µm diameter PLL-PEG-coated fluorescent beads (Fluorosphere #F8810, Life Technologies (Carlsbad, CA)) and sonicated before addition of APS and TEMED. 25 µL of this solution was added onto the micropatterned photomask, covered with a silanized coverslip, and allowed to polymerize for 25 min before being gently detached in sodium bicarbonate buffer. Micropatterns were stored overnight in sodium bicarbonate buffer at 4°C before plating cells. Non-patterned hydrogels were prepared using the same polyacrylamide polymerization procedure followed by a step of oxygen plasma treatment for 30 s at 30 W of the coverslips and subsequent incubation with fibronectin for 30 min.

For identification of untransfected MVE-KO cells on patterns, the cells were stained with 1 µM CellTracker Green (Invitrogen) for 5 min followed by removal of unbound dye with PBS. In case of reconstituted MVE-KO cells, only cells expressing EGFP-Evl that exhibited appropriate localization of the fusion protein at focal adhesions on the patterns were selected for performing traction force experiments. Images were acquired on an EclipseTi-E Nikon confocal spinning disk microscope equipped with a CSUX1-A1 Yokogawa confocal head.

Data were analyzed with a set of macros in Fiji (*Martiel et al., 2015*). Displacement fields were obtained from bead images prior and after removal of cells by trypsin treatment. Bead images were first paired and aligned to correct for experimental drift. Displacement field was calculated by particle imaging velocimetry (PIV) on the base of normalized cross-correlation following an iterative scheme. Erroneous vectors where discarded owing to their low correlation value and replaced by the median value of neighboring vectors. Traction-force field was subsequently reconstructed by Fourier Transform Traction Cytometry, with a regularization parameter set to $3.2 \times 10^{-10}$.

## Statistical analyses

Quantitative experiments were performed at least in triplicates to avoid any potential bias of environmental influences or unintentional error. Impact on lamellipodial or adhesion phenotypes derived from analyses on fixed samples and living cells were systematically obtained from sample sizes of dozens or hundreds of cells, respectively. Raw data were processed in Excel (Microsoft). Statistical analyses were performed using SigmaPlot 11.0 software (Systat Software, Erkrath, Germany) or GraphPad Prism 5 (GraphPad, San Diego, CA). All data sets were tested for normality by the Shapiro-Wilk test. Statistical differences between normally distributed datasets of two groups were determined by the t-test and not normally distributed datasets of two groups by the non-parametric Mann-Whitney U rank sum test. For comparison of more than two groups, statistical significance of normally distributed data was examined by one-way ANOVA and Tukey Multiple Comparison test. In case of not normally distributed data, the non-parametric Kruskal-Wallis test and Dunn's Multiple Comparison test were used. Statistical differences were defined as $*p \leq 0.05$, $**p \leq 0.01$, $***p \leq 0.001$ as well as n.s., not significant, and are displayed and mentioned in figures and figure legends, respectively. Graphs were created with Origin 2018G (OriginLab, Northampton, Ma), and final figures prepared with Photoshop (Adobe, San Jose, CA) and CorelDraw Graphics Suite 6X (CorelDraw, Ottawa, ON).

## Acknowledgements

This work was supported in part by the Deutsche Forschungsgemeinschaft (DFG), grants FA330/11-1 (to JF), RO2414/5-1 PROCOMPAS graduate program GRK2223/1 (to KR) as well as by ERC Advanced grants AAA 741773 (to LB) and CoG 724373 (to MS), and the Austrian Science Foundation (FWF) (to MS). We also thank Prof. Dr. em. Brigitte Jockusch and Sabine Buchmeier (PROCOMPAS monoclonal antibody facility, TU Braunschweig) for antibody generation, and the ANR-17-EURE-003 for sponsoring the live microscopy facilities of DBSCI department of IRIG (MuLife).

## Additional information

### Funding

| Funder | Grant reference number | Author |
|---|---|---|
| Deutsche Forschungsgemeinschaft | FA 330/11-1 | Jan Faix |
| H2020 European Research Council | AAA 741773 | Laurent Blanchoin |
| H2020 European Research Council | CoG 724373 | Michael Sixt |
| Deutsche Forschungsgemeinschaft | RO2414/5-1 | Klemens Rottner |
| Technische Universität Braunschweig | GRK2223/1 | Klemens Rottner |
| Austrian Science Fund | | Michael Sixt |

The funders had no role in study design, data collection and interpretation, or the decision to submit the work for publication.

### Author contributions

Julia Damiano-Guercio, Formal analysis, Validation, Investigation, Visualization, Writing - original draft; Laëtitia Kurzawa, Formal analysis, Validation, Investigation; Jan Mueller, Formal analysis, Investigation, Visualization; Georgi Dimchev, Maria Nemethova, Formal analysis, Investigation; Matthias Schaks, Data curation, Formal analysis, Investigation; Thomas Pokrant, Validation, Investigation; Stefan Brühmann, Formal analysis; Joern Linkner, Investigation; Laurent Blanchoin, Michael Sixt, Supervision; Klemens Rottner, Conceptualization, Data curation, Supervision, Funding acquisition,

Validation, Investigation, Writing - review and editing; Jan Faix, Conceptualization, Resources, Funding acquisition, Investigation, Visualization, Writing - original draft, Writing - review and editing

## Author ORCIDs
Julia Damiano-Guercio https://orcid.org/0000-0002-7948-9667
Georgi Dimchev https://orcid.org/0000-0001-8370-6161
Matthias Schaks https://orcid.org/0000-0001-6362-7909
Laurent Blanchoin https://orcid.org/0000-0001-8146-9254
Michael Sixt http://orcid.org/0000-0002-6620-9179
Klemens Rottner http://orcid.org/0000-0003-4244-4198
Jan Faix https://orcid.org/0000-0003-1803-9192

## Decision letter and Author response
Decision letter https://doi.org/10.7554/eLife.55351.sa1
Author response https://doi.org/10.7554/eLife.55351.sa2

---

# Additional files

## Supplementary files
- Supplementary file 1. Key resources table.

- Supplementary file 2. Sequences of generated knock out clones.

- Transparent reporting form

## Data availability
All data generated or analyzed during this study are included in the manuscript and supporting files. Source data files have been provided for all Figures.

---

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
