## [Decision Letter]

**Acceptance summary:**

Ena/VASP family proteins are central regulators of actin filament assembly, but their precise roles in cell migration, morphogenesis and adhesion have remained somewhat controversial. By generating and analyzing single, double, and triple Ena/Mena/VASP knockout cells, Damiano-Guercio et al. revealed here that Ena/VASP family proteins are positive regulators of migration in different cell-lines. Moreover, they demonstrate that Ena/VASP family proteins are critical for the formation of microspikes, but not for the formation of filopodia, thus providing evidence that microspikes arise exclusively through convergent elongation of lamellipodial actin filaments, whereas filopodia can be generated by different mechanisms. Finally, they provide evidence that Ena/VASP family proteins contribute to cell spreading and focal adhesion formation. Collectively, this study provides important new insights into the cellular functions of Ena/VASP family proteins.

**Decision letter after peer review:**

Thank you for submitting your article "Loss of Ena/VASP interferes with lamellipodium architecture, motility and integrin-dependent adhesion" for consideration by *eLife*. Your article has been reviewed by three peer reviewers, including Pekka Lappalainen as the Reviewing Editor and Reviewer #1, and the evaluation has been overseen by Anna Akhmanova as the Senior Editor. The following individual involved in review of your submission has agreed to reveal their identity: Giorgio Scita (Reviewer #2).

The reviewers have discussed the reviews with one another and the Reviewing Editor has drafted this decision to help you prepare a revised submission.

Ena/VASP family proteins promote actin polymerization at lamellipodia, filopodia, focal adhesions, and stress fibers in animal cells. However, whether Ena/VASP family proteins function as positive or negative regulators of cell migration has remained controversial. Moreover, the precise role of Ena/VASP family proteins in the formation of filopodia and microspikes, and the possible differences between the assembly-mechanisms of these two types of actin filament bundles, were elusive.

Damiano-Guercio et al. provide a careful and comprehensive analysis of the role of Ena, Mena and VASP in the motility of melanoma cells and fibroblasts through CRISPR/Cas-mediated loss-of-function experiments. They provide compelling evidence for the role of Ena/VASP family proteins in promoting cell migration, at variance with what was reported in some earlier publications. They further demonstrate that Ena/VASP family proteins are critical for the formation of microspikes within the lamellipodium, while filopodia can assemble also in the absence of Ena/VASPs. At the ultrastructural level, actin network was shown to be severely affected following removal of Eva/VASP family proteins. Finally, using the same controlled systems the authors show that focal adhesions are impaired in Evl/Mena/VASP-deficient cells. This manuscript therefore clarifies and extends some earlier and partially conflicting literature.

The experiments presented in the manuscript are of very good technical quality, and the study provides interesting new information on the roles of Ena/VASP family proteins in cell migration and microspike/filopodia formation. However, the reviewers felt that the novel conceptual insight of this study is, at least in its present form, somewhat limited. Therefore, additional experiments and better discussion of the data are required to increase the impact and novelty of the study.

Essential revisions:

1) The differential effects of Evl, Mena and VASP on cell migration and focal adhesions are interesting, and should be examined in more detail. Thus, the authors should carefully compare the localizations these proteins in lamellipodia and focal adhesions. This could be done either by antibody stainings of non-transfected cells, or by expressing Evl and VASP with different epitope tags in the same cells. Moreover, the authors should perform FRAP or photoactivation experiments to reveal whether Evl and VASP display differences in their dynamics at lamellipodia and/or focal adhesions. Finally, it is important to quantify the ectopic expression levels of these proteins in rescue cells. Knowing how "rescue" expression levels compare to endogenous control cells would very helpful to interpret these data.

2) The altered structural organization of the actin network in EVM-KO cells suggests that these cells might have a hard time to move particularly under conditions where the actin network has to oppose strong micro-environmental forces. Thus, it would be informative to test the migration of EVM-KO in 3D setting or against a flow, as these experiments might provide novel insights and broaden the relevance of Ena/VASP family proteins in cell migration to levels not previously tested.

3) Some important questions concerning the mechanism by which the loss of Ena/VASP leads to altered lamellipodial architecture and dynamics remain unanswered. Specifically, the increase of Arp2/3 and CP densities in lamellipodia assembling in the absence of Ena/VASP is difficult to comprehend. Any data the authors can provide in explaining this observation would strengthen the study. Alternatively, this issue could be better discussed in the manuscript.

4) The authors should rewrite the Introduction and Discussion to place their study in a better context with the previous work. For example, positive correlation of VASP accumulation at the leading edge with the protrusion rate of lamellipodia has been already observed at least in some cell types (e.g. Rottner et al., 1999, Lacayo et al., 2007). In addition, some Ena/VASP family members have been shown to promote cell motility and invasion in certain scenarios (e.g. Philippar et al., 2008). Finally, a wealth of biochemical work has convincingly established that Ena/VASP proteins possess actin polymerase and anti-capping activity (see Breitsprecher, 2008 and 2011, Mullins and Hansen, 2010). The authors' statement that Ena/VASP proteins "are mainly known as negative regulators of cell motility" appears to mainly rest on two high profile papers from the Bear/Gertler labs. By fixating on selected aspects of these early studies, the authors have created an unnecessary straw man argument.

---

## [Author Response]

Essential revisions:1) The differential effects of Evl, Mena and VASP on cell migration and focal adhesions are interesting, and should be examined in more detail. Thus, the authors should carefully compare the localizations these proteins in lamellipodia and focal adhesions. This could be done either by antibody stainings of non-transfected cells, or by expressing Evl and VASP with different epitope tags in the same cells. Moreover, the authors should perform FRAP or photoactivation experiments to reveal whether Evl and VASP display differences in their dynamics at lamellipodia and/or focal adhesions. Finally, it is important to quantify the ectopic expression levels of these proteins in rescue cells. Knowing how "rescue" expression levels compare to endogenous control cells would very helpful to interpret these data.

Despite the more prominent localization of Evl in focal adhesions and the more prominent localization of VASP at the leading edge as compared to the respective other in single transfection experiments, co‐transfections of EGFP‐tagged VASP with mCherry‐tagged Evl (or *vice versa*) in EVMKO cells was not adequate to report any convincing differences in localization between the two proteins. This can may explained by potential hetero‐oligomerization of ectopically expressed and tagged Ena/VASP proteins previously reported (Riquelme et al., 2015), likely interfering with the outcome in this specific experimental situation. However, as suggested, we have now also performed FRAP experiments of EGFP‐tagged VASP *versus* ‐Evl in focal adhesions and lamellipodia using rescued EVM‐KO cells. Notably, Evl shows a moderate, but statistically significant increase in the amount of immobile fraction in focal adhesions as compared to VASP, potentially explaining its more robust accumulation in these structures as compared to EGFP‐tagged VASP. No differences, however, between these proteins were observed for their turnover rates, neither in focal adhesions nor in lamellipodia, suggesting that the observed, functional relevance of Evl in focal adhesions may indeed derive from its increased stable association with specific focal adhesion components. This notwithstanding, VASP is also expressed at much higher levels as compared to Evl (Figure 2J), which by itself could explain its more prominent role in cell migration (these data are also now shown in Figure 7—figure supplement 2).

The last comment in this chapter refers to the unknown quantity of Ena/VASP family proteins in reconstituted EVM‐KO cells as compared to endogenous protein levels. We therefore quantified the average abundance of all three Ena/VASP members from immunoblots by densitometry in the entire cell population of rescued EVK‐KO cells. As now shown in Figure 1—figure supplement 1, in rescued cells, Evl was on average overexpressed 2.5‐fold, VASP 1.5‐fold and Mena 1.1‐fold as compared to B16‐F1 control. Nevertheless, since we usually use low or at best moderately expressing cells for our microscopy analyses and keeping in mind the high variability of expression levels of EGFP‐tagged proteins in our transfected cell populations, we assume amounts of respective Ena/VASP members in cells analyzed by microscopy to even be considerably lower than these values.

2) The altered structural organization of the actin network in EVM-KO cells suggests that these cells might have a hard time to move particularly under conditions where the actin network has to oppose strong micro-environmental forces. Thus, it would be informative to test the migration of EVM-KO in 3D setting or against a flow, as these experiments might provide novel insights and broaden the relevance of Ena/VASP family proteins in cell migration to levels not previously tested.

This is indeed a very interesting question, since Ena/VASP proteins have previously also been implicated in invasion of cancer cells (Philippar et al., 2008). Thus, as suggested, we analyzed migration of EVM‐KO and B16‐F1 control cells in Matrigel invasion assays. However, and to our surprise, as now shown in Figure 1—figure supplement 3, we did not find any significant differences in invasion rates between both cell lines. Although we cannot currently explain the mechanistic details of this result, which have to be unraveled in future studies, this result suggests that the robust, Ena/VASP‐driven promotion of actin‐based protrusion is of little importance for migration in confinement, which particularly relies on actomyosin contractility, as previously reported for other cell types (Poincloux et al., 2011; Ramalingam et al., 2015).

3) Some important questions concerning the mechanism by which the loss of Ena/VASP leads to altered lamellipodial architecture and dynamics remain unanswered. Specifically, the increase of Arp2/3 and CP densities in lamellipodia assembling in the absence of Ena/VASP is difficult to comprehend. Any data the authors can provide in explaining this observation would strengthen the study. Alternatively, this issue could be better discussed in the manuscript.

We think we can reasonably well explain the increase of Arp2/3 and CP densities in lamellipodia assembling after loss of Ena/VASP, but we apologize if our description was not sufficiently comprehensive for our readers. Ena/VASP proteins are actin polymerases thought to compete with CP for growing filament barbed ends (Bear et al., 2002; Breitsprecher et al., 2008, 2011; Hansen and Mullins, 2010). One possibility therefore is that the absence of Ena/VASP may allow for an increased proportion of CP localizing to filament barbed ends in the lamellipodium tip. An increased presence of CP might then be sufficient to increase the activity of Arp2/3 complex, as CP has also been reported to operate as co‐factor of Arp2/3‐dependent actin assembly (Akin and Mullins, 2008). Consistent with this, overexpression of VASP in B16‐F1 cells was previously found to decrease Arp2/3 complex intensities in lamellipodia (Dimchev et al., 2017). However, the latter finding could also speak for an alternative explanation for our observations. In fact, VASP‐mediated suppression of Arp2/3 complex incorporation into lamellipodia would also be consistent with Ena/VASP family members increasing the branch spacing of filaments in Arp2/3‐dependent actin tails, as previously observed in vitro (Pernier et al., 2016; Samarin et al., 2003). Thus, upon loss of Ena/VASP, filaments may not only become shorter, but Arp2/3 complex intensities simply increase due to enhanced branching activity. In this alternative possibility, increased Arp2/3 densities might in turn indirectly contribute to increased CP densities. The latter view is supported by the observation that acute, experimental sequestration of Arp2/3 complex (by microinjection of WCA) coincides with removal of both Arp2/3 complex and CP from lamellipodia, suggesting that capping protein accumulation in the lamellipodium does indeed depend on the presence of Arp2/3 complex (Koestler et al., 2013). Future experiments will have to define which one of these possibilities will actually hold true, or whether it might even be a combination of both. All these aspects will now be discussed in more detail in the revised version of our manuscript.

4) The authors should rewrite the Introduction and Discussion to place their study in a better context with the previous work. For example, positive correlation of VASP accumulation at the leading edge with the protrusion rate of lamellipodia has been already observed at least in some cell types (e.g. Rottner et al., 1999, Lacayo et al., 2007). In addition, some Ena/VASP family members have been shown to promote cell motility and invasion in certain scenarios (e.g. Philippar et al., 2008). Finally, a wealth of biochemical work has convincingly established that Ena/VASP proteins possess actin polymerase and anti-capping activity (see Breitsprecher, 2008 and 2011, Mullins and Hansen, 2010). The authors' statement that Ena/VASP proteins "are mainly known as negative regulators of cell motility" appears to mainly rest on two high profile papers from the Bear/Gertler labs. By fixating on selected aspects of these early studies, the authors have created an unnecessary straw man argument.

We apologize for focusing too much on the two papers by Bear/Gertler labs published in Cell (Bear et al., 2002, 2000). However, one should keep in mind that these papers, which are still highly cited, were outstandingly influential for the interpretation of Ena/VASP activity in cells, and not only that, they also served as benchmarks for the motility field in general, and this for two decades now. It is therefore not appropriate to accuse us of having generated something that the reviewer terms strawman argument, this is simply not fair. We should also emphasize that as scientists, we believe it must be legitimate and necessary to critically re‐evaluate even classic, high‐ranking scientific publications, in particular in modern times having access now to exciting, novel experimental tools such as CRISPR/Cas9. The reviewer will certainly agree that these approaches now allow us to confirm or to correct previous concepts and models that have been using on technology available at the time. Having said this, however, we do apologize if other relevant publications have fallen too short in this particular context! Indeed, we have now extended our text in this direction, as requested, and significantly rewritten both Introduction and Discussion to place our study in better context with previous work.